# Assessment of the Suitability of Non-Air-Conditioned Historical Buildings for Artwork Conservation: Comparing the Microclimate Monitoring in Vasari Corridor and La Specola Museum in Florence

Fabio Sciurpi *, Cristina Carletti, Gianfranco Cellai and Cristina Piselli

Department of Architecture DIDA, University of Florence, 50121 Florence, Italy
* Correspondence: fabio.sciurpi@unifi.it

**Abstract:** The current energy crisis and the necessity to minimize energy waste suggest the need to assess non-air-conditioned buildings in terms of the need to install an air-conditioning system and to size and control it efficiently. This applies to historical museum buildings hosting artworks that require specific microclimate conditions for their preservation. With this view, this work analyzes the suitability of non-air-conditioned historical museum buildings to properly preserve exhibits. Therefore, two non-air-conditioned museums located in the historical city center of Florence, Italy, are considered as case studies, i.e., Vasari Corridor and La Specola. One year of indoor microclimate data monitored in representative rooms of the museums are analyzed according to the standard for artworks preservation and in terms of historical climate. Results of monitored indoor air temperature and relative humidity show that all monitored rooms are not suitable for the preservation of the exhibits without the installation of an air-conditioning system. However, to minimize the energy consumption, the hygrothermal control can be based on the observed historical climate that characterizes the environments, which presents acceptable preservation ranges much wider that the reference technical standard. In this way, the energy needs for the environmental control necessary to ensure the good conservation of the artworks can be significantly reduced.

**Keywords:** museum; environmental monitoring; artwork conservation; historical buildings; historical climate; energy efficiency; climate adaptation

## 1. Introduction

Over the last 40 years, the concept of museum has changed substantially, and the definition as a "static place", intended to preserve, protect, conserve, and display the cultural heritage, has been replaced by a "dynamic vision" of transmission and dissemination of knowledge. Currently, the museum is conceived of as a permanent institution open to the public with the aim of education, study, and enjoyment [1]. Moreover, museums play a key role in society, not only for their cultural value but also due to their touristic interest. As the increasing cultural tourism can also be a way to achieve financial sustainability, they embody a priceless economic value. Hence, it is necessary to ensure and improve the exhibits' conservation conditions, as well as the wellbeing and livability of visitors and staff [2].

The main possible causes for the deterioration of artworks were summarized by ICCROM (International Center for the Study of the Preservation and Restoration of Cultural Property, Rome, Italy) [3] as physical forces, thieves and vandals, fire, water, pets, pollutants, light, temperature, relative humidity, and dissociation. With regard to the environmental parameters, the stability of indoor temperature and relative humidity is a key player for exhibit preservation and thermal comfort in museums [4]. Therefore, the hygrothermal conditions inside museums have to be carefully monitored and assessed [5]. Moreover, the

fluctuations of indoor temperature and relative humidity can cause changes in the moisture equilibrium of the organic hygroscopic materials that can lead to important biological and mechanical degradation phenomena [6].

Preservation is the action taken to retard or prevent deterioration or damage to cultural heritage by controlling the environment and/or by treating the structure to maintain it as best as possible in an unchanging state [7]. To preserve exposed artworks or artifacts and to reduce the process of exhibits degradation, preventive conservation strategies are fundamental in museums [8,9]. These actions have to be adopted for the prevention and, mostly, the control of environmental parameters by taking into account the different aspects of the museum involved: building envelope, collections features, and visitors. Therefore, preventive conservation implies both passive techniques, aimed at minimizing the potential damage to the object by the exhibition environment, and environmental monitoring and control strategies, aimed at minimizing the fluctuations of indoor environmental parameters and reducing the impacts of outdoor conditions (e.g., windows openings, visitors crowding, irregular HVAC (heating, ventilation, and air conditioning) system operation, or wrong positioning of HVAC terminals and lighting elements) [10]. The management of museums is difficult especially when they are placed in historical buildings, where the refurbishment and the conversion into exhibition spaces according to a "modern vision" usually involve several conservation risks.

Additionally, the needs for artwork conservation and human comfort are often difficult to combine and maintain; human comfort requires specific levels of light, temperature, and relative humidity which are not always compatible with the safeguarding of the exhibits [11]. Moreover, as for thermal comfort, the expectancy of museum visitors is different from that of the staff. The reasons could be several; for instance, museum visits are usually shorter than a workday and visitors generally move around the museum, while the staff are often static. Furthermore, visitors' clothing insulation level, their activity, and their pattern of use can be greatly different from those of the staff [12].

The degradation issues and the microclimatic parameters to be guaranteed in the museum are reported in different technical standards that help curators, indicating the appropriate ranges of temperature and relative humidity to be maintained. However, in some cases, when the artifact has been acclimatized for years to specific historical climate conditions, it is advisable to maintain these conditions instead of standard ones [13]. Indeed, if the artwork has been adapted to these conditions, a variation could involve climatic shocks the artwork [14]. Italian technical regulations and guidelines about cultural heritage conservation, such as UNI 10829:1999 [15] and D.M. 10.05.2001 [16], list optimal and acceptable ranges for temperature and relative humidity, as well as establish guidelines and methods to measure indoor temperature and relative humidity values. These values are generally more conservative than the values that can be deduced from the historical climate. The European standard UNI EN 15757:2010 [17] defines the historical climate as the "climatic conditions in a microenvironment where a cultural heritage object has always been kept or has been kept for a long period of time (at least 1 year), and to which it has become acclimatized". The ASHRAE Standard [18] and the standard UNI EN 15757:2010 [17] illustrate specifications for temperature and relative humidity to limit climate-induced mechanical damage in organic hygroscopic materials and proper monitoring strategies. In particular, UNI EN 15757:2010 [17] constitutes a guideline that specifies the levels of indoor temperature and relative humidity to limit the physical damage that the climate causes to hygroscopic organic materials that have been kept stored or in exhibitions in indoor environments, such as museums, for a long time (at least over 1 year).

To assess the museum environment's suitability to conserve exhibits and guarantee users' comfort, action plans are required to reduce degradation risk and to develop preventive control programs aimed at maintaining the optimal microclimatic conditions, involving the microclimatic monitoring and analyses, as well as the construction of a digital twin building energy model (BEM) [19,20]. Proper analysis of data and dynamic simulation of the museum environment are fundamental to simulate the thermal behavior of the

building. Indeed, monitoring and simulation allow conservators to create useful databases for the assessment of the performance of the museum and the evaluation of the influence of different retrofitting strategies [21,22]. Corgnati and Filippi [23] proposed an operational procedure to assess and define the thermo-hygrometric quality level in exhibition areas of large museums through a synthetic index (PI) for the assessment of prevention risk based on microclimate monitoring. On the other hand, Ishikawa et al. [24] implemented a simulation model of a museum storage room to evaluate the cause and potential strategies to face unacceptable humidity conditions for the preservation of artifacts.

Italy has a large collection of cultural heritage, with over 4000 museums [25]. Cultural heritage buildings are extremely energy-consuming; evaluations made by the Italian National Agency for New Technologies, Energy, and Sustainable Economic Development (ENEA) showed that the national museum sector requires an annual energy expenditure of approximately 250 million EUR, and consumption has increased by 50% compared to the 1980s [26]. In particular, air conditioning and lighting weigh the most on the energy needs of Italian museums, which make them among the Public Administration buildings with the highest environmental impact [26]. Their energy bills fluctuate on average from 780 to 1280 GW per year, a considerable expenditure that, in some cases, reaches up to 70% of the budget of the museum institute [26]. Therefore, strategies to reduce this energy impact are extremely needed. Furthermore, in the last few years, the cost of energy has undergone significant variations. If, in 2020, the year characterized by the pandemic phenomenon COVID-19, the global economy received a strong setback, 2021 was, instead, the year of recovery. However, it was accompanied by a strong increase in prices, which for natural gas reached more than 100 EUR/MWh, i.e., more than 15 times higher than the values recorded in mid-2020 [27]. This already high value, due to the war in Ukraine, peaked at around 300 EUR/MWh at the end of August 2022 [27]. Considering the Italian electricity production mix, which still sees a significant use of gas, the gas price increase inevitably reflected on electricity prices, with the national price reaching completely unprecedented values, frequently exceeding the threshold of 300 EUR/MWh, in 2021, and 600 EUR/MWh at the end of August 2022 [27]. This process is inevitably impacting the final consumers, despite the measures taken by the Italian government (e.g., excise duty reduction, VAT reduction, etc.) to contain the weight of the increase in wholesale prices on citizens [28].

In this panorama, this work proposes an approach for the energy efficient operation of heritage museum buildings. The study takes the lead from previous works that analyzed the microclimatic data monitored in two non-air-conditioned historical buildings in Florence, Italy, i.e., the Vasari Corridor [29,30] and La Specola museum [31–33]. The aim of this study was first to compare the data monitored in the two buildings to characterize and assess the responses in terms of indoor microclimate of non-air-conditioned buildings to outdoor and indoor variations. In particular, the monitored data are analyzed considering the optimal values suggested by the standards for the conservation of artworks [15,16] and the values deriving from the historical climate that characterizes the analyzed environments [17]. Therefore, the final goal is to verify the suitability of non-air-conditioned buildings to properly preserve hygroscopic artworks thanks to the capability of adaptation to historical climate conditions. Indeed, although there are studies in the literature focused on historical climate conditions in air-conditioned buildings [34–36], to the best of the authors' knowledge, free-running exhibition spaces are poorly addressed by the existing literature. Indeed, the high energy impact and the level reached by energy costs is jeopardizing the environmental and economic sustainability of HVAC systems in museum buildings. In this view, this approach represents a strategy to minimize the use of HVAC systems in heritage museum buildings to achieve energy and economic savings.

## 2. Materials and Methods

### 2.1. Methods

The steps of the research study are as follows:

(i.) Long-term indoor microclimate monitoring in non-air-conditioned heritage museum buildings. Preliminary monitored data were already presented in previous studies [29–33]; the main outcomes are also reported in this work for the sake of comprehensiveness and comparison.

(ii.) Comparison between the microclimate conditions monitored in the different considered buildings.

(iii.) Analysis of the hygrothermal conditions and the climate adaptation in the heritage museum buildings to evaluate how construction characteristics influence the performance of the building and if non-air-conditioned historic building museums are suitable to properly preserve hygroscopic artworks.

### 2.2. Case Studies: Palazzo Vecchio–Palazzo Torrigiani Elevated Path

The elevated external passage known as the Vasari Corridor is an enclosed and privileged connection built in 1565 to join Palazzo Vecchio with Palazzo Pitti, in the historical city center of Florence, Italy. The scientific museum La Specola is located in Palazzo Torrigiani (16th century), located in the same area of the city center. In the 19th century, a corridor was built connecting Palazzo Pitti with Palazzo Torrigiani, thus creating an entirely elevated and enclosed path that connects Palazzo Vecchio with Palazzo Torrigiani, in a configuration that is unique in the world (Figure 1).

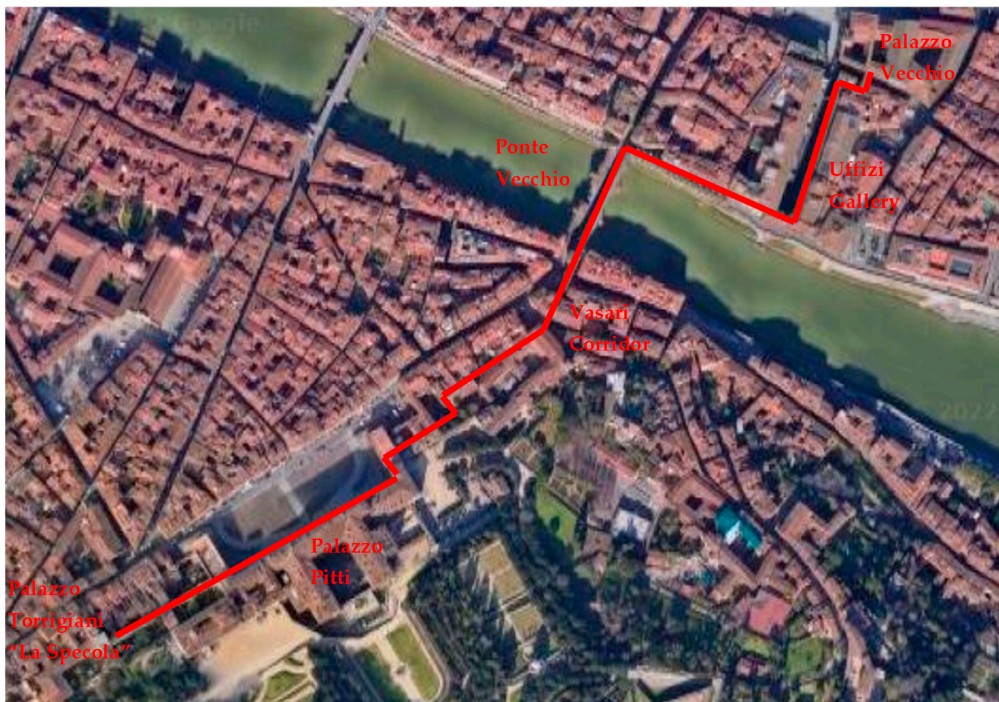

**Figure 1.** View of the path that connects Palazzo Vecchio with Palazzo Torrigiani.

Two historical museum buildings are analyzed in this study: a section of the Vasari Corridor and some rooms in La Specola museum, as described below.

### 2.2.1. Vasari Corridor

The Vasari Corridor was commissioned by Duke Cosimo I de' Medici to the architect Giorgio Vasari in the occasion of the wedding between his eldest son, Prince Regent Francesco, and Archduchess Joanna of Austria, sister of Emperor Maximilian II, celebrated on 18 December 1565. The Corridor starts from Palazzo Vecchio, the heart of the ancient Florentine political government, enters inside the Uffizi Gallery, runs along Lungarno degli Archibusieri, crosses the river Arno by passing over Ponte Vecchio, then enters Boboli Garden, and leads inside Palazzo Pitti. The total path is about 760 m, and its width ranges

from 1.3 m to 4 m, with the height varying from 3.4 m to 4.3 m. According to the original intentions of Cosimo I de' Medici and Giorgio Vasari, the corridor was meant for a private and exclusive use allowing the duke to move from the government seat of Palazzo Vecchio to their residence of Palazzo Pitti in total safety. After three centuries of such a private use, it was opened to the public in 1866. Seriously damaged by mines placed by the retreating German army during the Second World War, the corridor was restored and partly rebuilt in the 1950s. Owing to the flood of the river Arno that occurred on 4 November 1966, the corridor suffered many structural problems and was again restored to be reopened in 1973. Since 2017, the Vasari Corridor has been closed to visitors for refurbishment.

The artworks exhibited in the Vasari Corridor are mostly paintings on canvas and oil paints on canvas (mainly self-portraits) from different historical periods, which hang on the walls [29,30].

From the constructive and orientation point of view, the corridor can be divided into three sections [30]:

- Lungarno degli Archibusieri (hereinafter referred to as Lungarno) ranging from the Uffizi to Ponte Vecchio, parallel to the river Arno and with small windows and a predominant southern orientation (geometric and construction details are reported in Tables 1 and 2);
- Ponte Vecchio, which crosses the river Arno with east–west orientation, characterized in the central part by the presence of large windows (geometric and construction details are reported in Tables 1 and 2); The final part of the corridor, passing through the Florentine historical buildings, with few and small windows, being more protected from outdoor atmospheric conditions.

**Table 1.** Geometric data of the Ponte Vecchio and Lungarno thermal zones of Vasari Corridor.

| Parameter | Ponte Vecchio | Lungarno |
|---|---|---|
| External surface * (S) [m$^2$] | 860.90 | 1315.60 |
| External volume (V) [m$^3$] | 1854.20 | 2149.10 |
| Ratio S/V [m$^{-1}$] | 0.46 | 0.61 |
| Window surface ($S_w$) [m$^2$] | 74.80 | 39.30 |
| Net floor surface ($A_f$) [m$^2$] | 469.40 | 420.50 |
| Ratio $S_w/A_f$ [-] | 0.16 | 0.09 |
| Ratio $S_w/S$ [-] | 0.09 | 0.03 |

* It does not include surfaces toward other adjacent rooms.

**Table 2.** Thermal and optical properties of the existing building envelope of Vasari Corridor.

| Building Component | Materials | U * [W/m$^2$K] | Yie ** [W/m$^2$K] | g *** [-] | $\tau_v$ **** [-] |
|---|---|---|---|---|---|
| External wall | Internal plaster, solid brick masonry, external plaster | 1.65 | 0.398 | - | - |
| Pitched wooden roof | Truss, double layer of joists, cast concrete, bitumen sheet, tile covering | 1.88 | 1.601 | - | - |
| External floor over vaults brick | Ceramic tiles, cement and sand screed, brick vault, plaster | 1.32 | 0.469 | - | - |
| Double window | Wood frame, single glazing | 2.09 | - | 0.69 | 0.74 |
| Single window | Wood frame, single glazing | 5.78 | - | 0.82 | 0.88 |

* Thermal transmittance; ** periodic thermal transmittance; *** solar factor; **** light transmittance.

In this study, the sections of Lungarno and Ponte Vecchio were selected and analyzed as representative non-air-conditioned buildings.

Figure 2 shows the plan (a) and a view (b) of the building. In the plan, the analyzed sections (Lungarno and Ponte Vecchio) are highlighted.

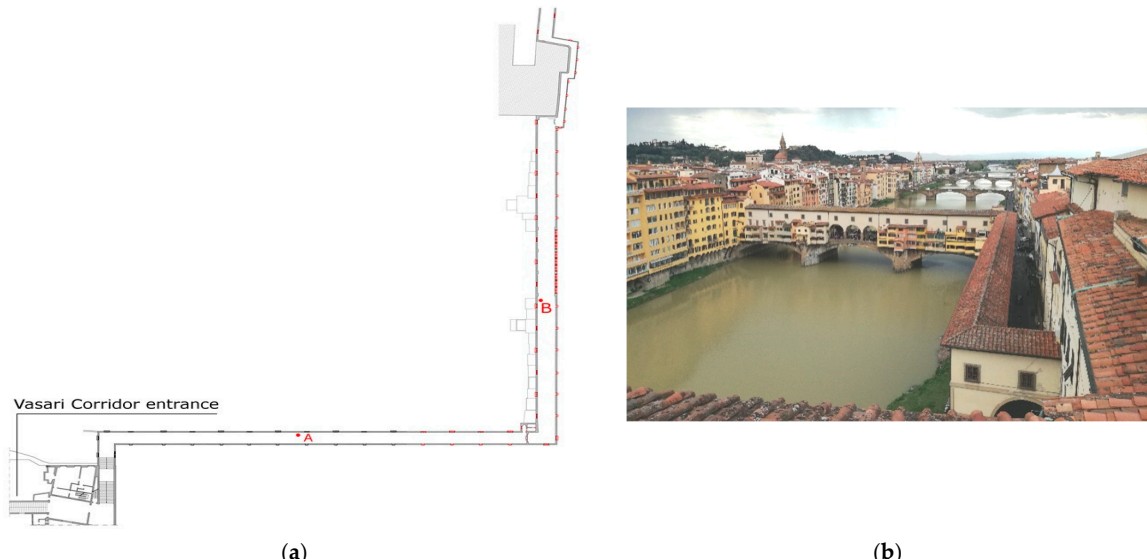

| (a) | (b) |

**Figure 2.** (**a**) Plan with the monitored zones (position A—Lungarno, and position B—Ponte Vecchio) highlighted; (**b**) a view of Vasari Corridor.

The building envelope of the corridor is made of poor materials: bricks and stones for the external walls, recycled from destroyed buildings or found in the Florentine plain or extracted from the Arno, and wood for the roof. There are three kinds of windows: double windows (composed of an inner and outer frame; both windows can be opened) made with wooden frame and single glazing with a roller shading between the two windows, double windows made with wooden frame and single glazing without shading, and single windows made with wooden frame with single glazing. Table 1 summarizes the main geometric data of thermal zones Ponte Vecchio and Lungarno. Given their initial end-use (i.e., corridor), these zones are distributed in a narrow and long area. Table 2 reports the thermophysical properties of the opaque and transparent envelope components (external walls, floors, roof, and windows), defined by local inspection. The knowledge of the thermophysical properties of the building envelope is necessary to understand the relationship between the variation of indoor and outdoor microclimate conditions in the unconditioned museum. Moreover, the picture of the test performed on the external wall depicted in Figure 3a shows that the building structure has remained substantially unchanged since the construction period.

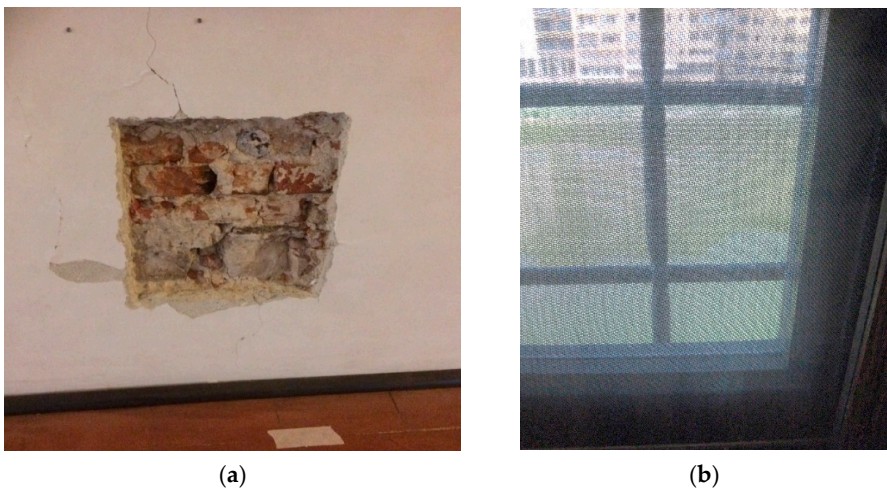

| (a) | (b) |

**Figure 3.** (**a**) View of the test performed on the external wall and (**b**) the roller shading on the external windows of the Vasari Corridor.

The windows with internal roller shading use a diffusing shades material, while the effect of the external grille on windows can be considered negligible in terms of transmission of solar radiation (Figure 3b). Window shading is controlled manually by adjusting the technical roller blind with semitransparent fabric, which allows visual perception of the exterior and the control of natural light and external thermal loads due to solar radiation.

As for the technical systems, the zones of Vasari Corridor are equipped with an artificial lighting system with fluorescent lamps, emergency signals, the anti-intrusion system, the video-control system, and a smoke detector. As previously mentioned, no HVAC system is in operation.

### 2.2.2. La Specola Museum

As mentioned above, La Specola museum is located at the end of the Palazzo Vecchio–Palazzo Torrigiani elevated path. It was inaugurated on 1775 thanks to the Grand Duke of Tuscany Pietro Leopoldo di Lorena, who reorganized the collections of the Medici family. Since 2019, the museum has been closed to the visitors for refurbishment.

The museum is divided in two sections: the zoological collection (rooms I–XXIV), with objects (especially taxidermic works) dated between the second half of the 19th century and the first decades of the 20th century, and the collection of anatomical waxes (rooms XXIV–XXXI), with objects (especially specimens of the human body made out of a mixture of waxes, resins, and dyes) dated between the second half of the 18th century and the second half of the 19th century. All the objects are exposed in the rooms and housed inside showcases of artistic and historic value, made of solid wood and single glazing dated to the first half of the 19th century [31–33].

Figure 4 depicts the plan (a) and a view (b) of La Specola museum. In the plan, the rooms (X and XXII, in the zoological section) selected and analyzed in this study are pointed out.

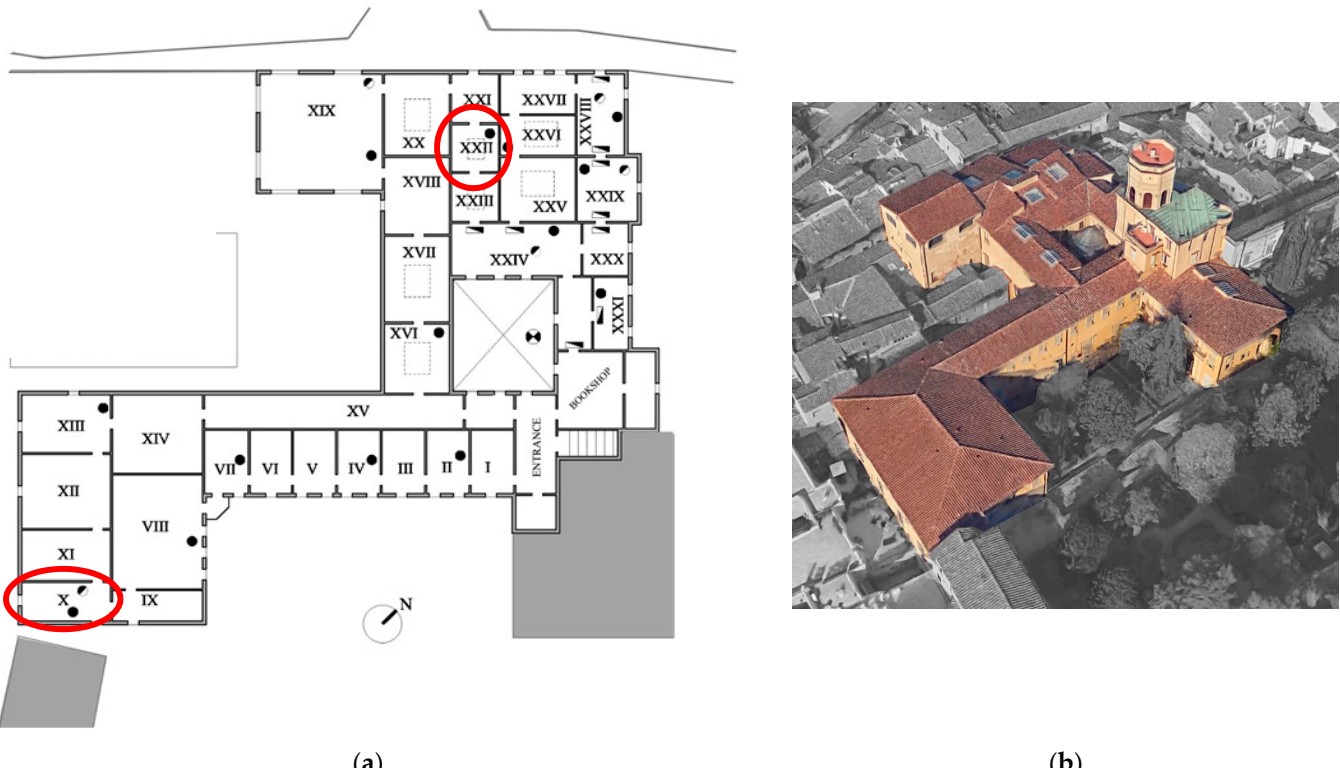

(**a**)　　　　　　　　　　　　　　　　　　　　　　(**b**)

**Figure 4.** (**a**) Plan with the monitored zones (rooms X and XXII) pointed out and (**b**) view of La Specola museum.

The building envelope of the museum is made of plastered stone and brick masonry external walls, wooden floors and roof, and false ceilings. Windows are made with a wood frame and single glazing, while skylights are made with a metal frame and single glazing. Shading devices, which are always closed, consist of external and internal shutters for the windows, while the skylights are equipped with internal drapes.

The zoological section is non-air-conditioned, whereas a heat pump system is present only in some rooms of the anatomical waxes section (rooms XXIV, XXVIII, and XXIX–XXXI), with ceiling fan coils that maintain the indoor air temperature between 20 °C and 22 °C [33]. The rooms of the museum are lit with tubular fluorescent lamps (turned on only during opening hours), while only some of the showcases are lit internally with tubular fluorescent or LED lamps.

According to the detailed results obtained in the previous studies [31–33], on the basis of specific criteria (i.e., trend of indoor microclimate parameters, air-conditioned or non-air-conditioned rooms, presence of windows or skylights, type of solar shadings, state of conservation and technical features of the external walls, conservative requirements of the exhibits, and type and relevance of the exhibits), the selected rooms X and XXII are considered representative of the non-air-conditioned area of the museum. Room X, located in the southwest corner of the building, has two external walls, one window with external and internal solar shading, and an uninsulated pitched roof. Room XXII, located in the northwest area of the building, is an internal room without external walls and is, thus, more protected from the outdoors; it also has one skylight without external solar shading and an uninsulated pitched roof. Moreover, both rooms have a much smaller area than the zones analyzed in Vasari Corridor. In room X, various mammals coming from the aquatic environment are exhibited, while, in room XXII, marine reptiles are displayed.

Table 3 shows the main geometric data of the museum envelope, relative to rooms X and XXII, and Table 4 reports the thermophysical properties of the opaque and transparent envelope components (external walls, floors, roof, and windows), defined by visual inspection and gathered from documentation on historical construction materials and techniques.

**Table 3.** Geometric data of La Specola museum rooms X and XXII.

| Parameter | Room X | Room XXII |
|---|---|---|
| External surface * (S) [m$^2$] | 75.70 | 41.90 |
| External volume (V) [m$^3$] | 261.00 | 217.00 |
| Ratio S/V [m$^{-1}$] | 0.29 | 0.19 |
| Window/skylight surface (S$_w$) [m$^2$] | 3.10 | 5.90 |
| Net floor surface (A$_f$) [m$^2$] | 67.00 | 40.30 |
| Ratio S$_w$/A$_f$ [-] | 0.05 | 0.15 |
| Ratio S$_w$/S [-] | 0.04 | 0.14 |

* It does not include surfaces toward other adjacent rooms.

**Table 4.** Thermal and optical properties of the existing building envelope of La Specola.

| Building Component | Materials | U * [W/m$^2$K] | Yie ** [W/m$^2$K] | g *** [-] | τ$_v$ **** [-] |
|---|---|---|---|---|---|
| External wall | Internal plaster, stone blocks, concrete filling, stone blocks, external plaster | 0.95–1.53 | 0.043–0.008 | - | - |
| Roof | Wood rafters, brick tiles, roof tiles | 2.99 | 1.960 | - | - |
| False ceiling | Lime plaster, reed trellis | 3.42 | - | - | - |
| Window | Wood frame, single glazing | 4.96 | - | 0.86 | 0.90 |
| Skylight | Metal frame without thermal break, single glazing | 5.83 | - | 0.86 | 0.90 |

* Thermal transmittance; ** periodic thermal transmittance; *** solar factor; **** light transmittance.

### 2.3. Monitoring Campaign

2.3.1. The Weather Data

For the purpose of the annual analysis of the case study buildings, the external hourly values of dry-bulb air temperature and relative humidity were monitored. For the monitoring campaign in Vasari Corridor, the data were provided for the whole year 2017 by the ancient Ximeniano astronomical observatory weather station located inside the city center of Florence [37] in close proximity of the corridor, while, for the campaign in La Specola museum, the data were recorded for 1 year between May 2012 and April 2013 with sensors (Tinytag Plus 2-TGP-4500) installed specifically in the courtyard of the building in a position shielded from direct solar radiation [33].

Figure 1 shows the positions of the two buildings within the city center of Florence; the maximum distance as the crow flies is about 300 m.

Figure 5 reports the monthly average values of dry-bulb air temperature (T) and relative humidity (RH) monitored in the two periods. The graphs show how the trend of T and RH values in the two periods, despite the different year and the origin of the data, are definitely comparable, especially after March. The small differences are partly attributable to the position of the sensor in the inner courtyard of La Specola less exposed to natural ventilation, where the RH values tend to remain constantly higher, while T values are slightly higher. In both cases, the temperatures reach the maximum values in the months of July–August and the minimum values in January. Although the variation of RH is flatter, the maximum and minimum values are inversely proportional to the trend of temperatures.

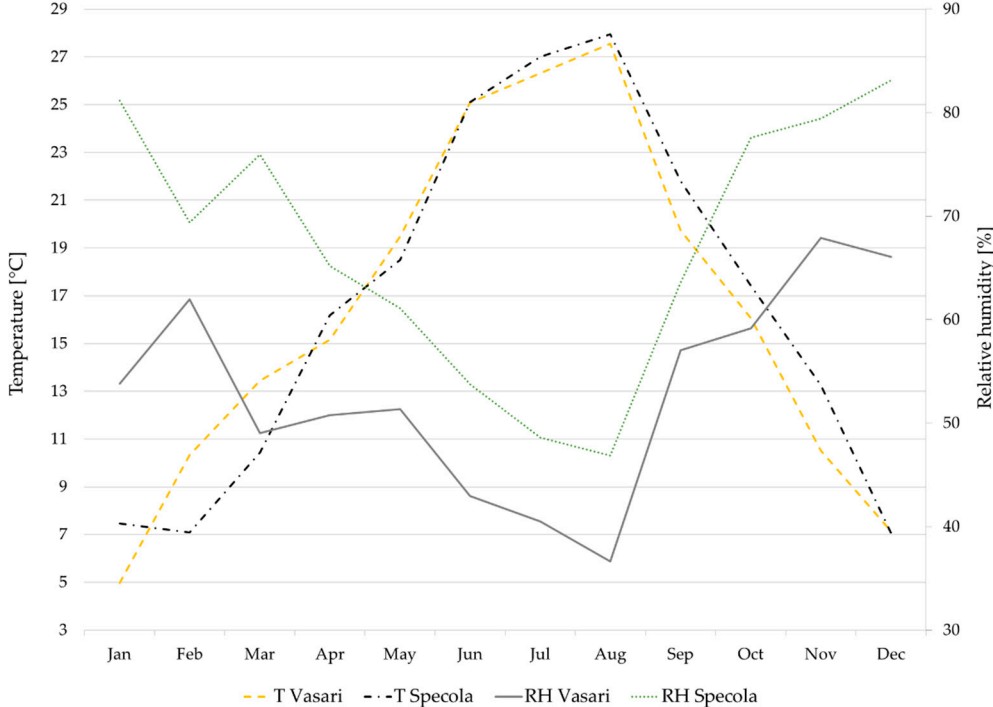

**Figure 5.** Vasari Corridor and La Specola: trend of monthly average outdoor air temperature (T) and relative humidity (RH).

In non-air-conditioned buildings, the external hygrothermal conditions significantly affect the indoor microclimate, defined by average levels and variability of T and RH. The more the building envelope components are characterized by high thermal transmittance and low thermal inertia and transparent surfaces exposed to solar radiation, the higher the variability of indoor microclimate parameters is, as analyzed in the monitoring campaign.

2.3.2. Indoor Microclimate Monitoring Campaign of Vasari Corridor

To investigate the thermal and hygrometric conditions inside the Vasari Corridor, a continuous monitoring system was installed in July 2016. The monitoring setup, described in detail in [30], consists of a set of data loggers that measure dry-bulb air temperature (T) and relative humidity (RH) [38]. These types of sensors respect the uncertainties defined in the European standards UNI EN 15758:2010 [39] for temperature and UNI EN 16242:2013 [40] for relative humidity.

In this study, data monitored throughout 2017 for the zones Lungarno and Ponte Vecchio are analyzed and discussed.

According to the procedure for the measurement of microclimate parameters suggested by UNI 10829:1999 [15], a preliminary survey was carried out in each zone using portable devices for measuring spot T and RH in the nodes of a horizontal grid with sides of 1.5 m at a height of 1.5 m from the floor. The spot measurements were made in the standard conditions of the environments, in a time interval not greater than 1 h for each zone. On the basis of the results obtained in this preliminary phase and taking into account the functional needs of the zones (i.e., aspects related to the use of the environments), the operating requirements of the monitoring systems (avoiding proximity to heat sources, direct light, and direct contact with local disturbance causes that may affect their proper operation), and the geometric characteristics of the zones, the positions of the continuous monitoring system were identified. Figure 6 depicts the selected monitoring positions, named A (Lungarno zone) and B (Ponte Vecchio zone). The sensors continuously monitored the indoor microclimate parameters (T and RH) in the two zones every 15 min.

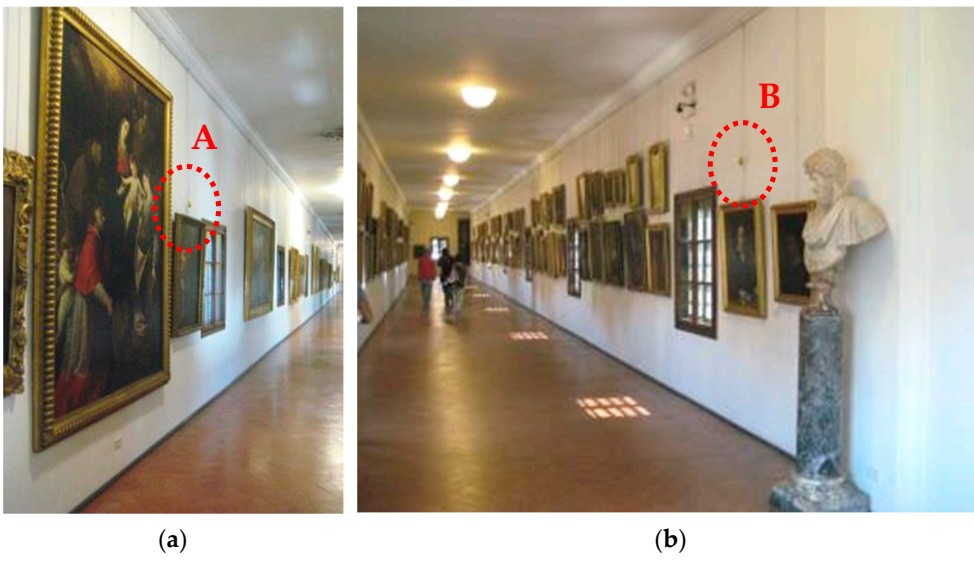

(**a**)                                     (**b**)

**Figure 6.** (**a**) Pictures of the dataloggers in positions A (Lungarno) and (**b**) B (Ponte Vecchio).

2.3.3. Indoor Microclimate Monitoring Campaign of La Specola Museum

To investigate the thermal and hygrometric conditions inside La Specola museum, a continuous monitoring system was installed in 2011. The details of the monitoring campaign are reported in previous studies [31–33] and summarized below for the purpose of the comparison with Vasari Corridor. Similarly to the other case study, the monitoring setup consists of a set of data loggers that measure dry-bulb air temperature (T) and relative humidity (RH) [38], which respect the requirement of the reference standards [39,40].

In this study, the data monitored in the rooms X and XXII between May 2012 and April 2013 were analyzed. These zones, without HVAC system, were selected as representative of a group of rooms in the museum. The sensor positions in the two zones are reported in Figure 7. The positions of the continuous monitoring system were defined according to the same procedure described for the Vasari Corridor (see Section 2.3.2). The sensors

continuously monitored the indoor microclimate parameters (T and RH) in the two zones every 15 min.

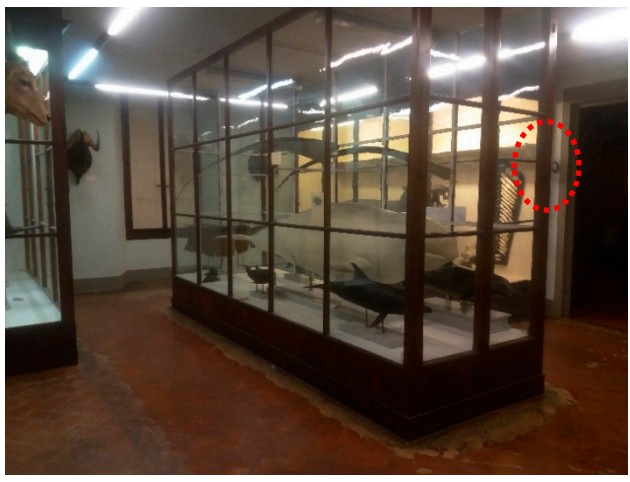

(**a**)

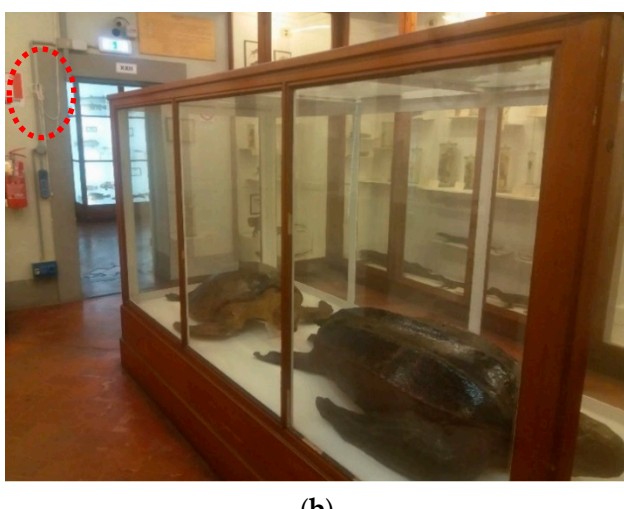

(**b**)

**Figure 7.** (**a**) Pictures of the dataloggers in room X and (**b**) room XXII.

*2.4. Data Analysis*

The monitored data were statistically analyzed and compared in terms of the following:

- Average values over the selected period, i.e., monthly and yearly;
- Hourly values for defining the statistical limits included in the 93rd and seventh percentiles, to exclude the largest and riskiest fluctuations;
- Short-term fluctuations, i.e., 24 h.

In addition, the trends of T and RH values were evaluated and shown in the graphics of the "performance index" (PI) for each zone. This index allows the assessment of the indoor microclimatic conditions in relation to the conservation capability of the object exhibited inside the zone. Indeed, it represents the percentage of time during which the evaluated microclimate parameters fall within the "acceptable" ranges [41]. Therefore, a higher PI value denotes better indoor environmental conditions for the conservation of the exhibition.

The abovementioned postprocessed data were used to verify if the climate had undergone any change and, in such a case, to what extent, to evaluate the suitability to preserve artworks and collection objects in non-air-conditioned buildings and which buildings characteristics mostly support this solution. The results obtained in the two case study buildings are discussed and compared. The analysis is carried out according to the main Italian reference standards D.M. 10.05.2001 [16] and UNI 10829:1999 [15]. The applicability of the latter, however, has been questioned in the existing literature [42]. Therefore, the data were also analyzed according to the standard UNI EN 15757:2010 [17], which applies a different approach based on methodological indications derived from the scrupulous analysis of the conservation environment [42].

As for the conservation, optimal microclimate parameters can be defined both by the curators and by technical documents and standards. These optimal values should be defined on the basis of the climate history of the exhibits and their materials and structural characteristics. Table 5 lists the reference values recommended by the Italian D.M. 10.05.2001 [16] for the conservation of the category of objects exhibited in the Vasari Corridor, i.e., "paintings on canvas, oil paints on canvas", and in the analyzed two rooms of La Specola museum, i.e., "organic material in general", as concerns absolute values (T and RH) and daily gradients ($\Delta T_{24}$ and $\Delta RH_{24}$) of temperature and relative humidity.

**Table 5.** Reference values recommended by the Italian D.M. 10.05.2001 [16] for the conservation of "paintings on canvas, oil paints on canvas" and "organic material in general".

| Type of Artefact | T [°C] | $\Delta T_{24}$ [°C] | RH [%] | $\Delta RH_{24}$ [%] |
|---|---|---|---|---|
| "Paintings on canvas, oil paints on canvas" | 19 ÷ 24 | ≤1.5 | 35 ÷ 50 | ≤6 |
| "Organic material in general" | 19 ÷ 24 | ≤1.5 | 50 ÷ 65 | ≤5 |

Moreover, it should be considered that, in accordance with the standard UNI EN 15757:2010 [17], for the conservation of artworks or exhibits involving hygroscopic materials, their historical climate should be prioritized. Indeed, the climate adaptation of the object to the temperature and relative humidity conditions, often for a long time (centuries), must take into account the storage history and the response of the object into the historical microclimate. UNI EN 15757 [17] recommends the historical climate to be maintained, especially as far as relative humidity is concerned, if the object has been found to be stable. In accordance, consistent values should be used as targets when controlling the microclimate conditions in the zone where these objects are conserved through HVAC systems or other strategies. Thanks to the microclimate monitoring for at least 1 year in both buildings, it was possible to define the historical climate for both museums, as detailed in Section 3, and use it to propose energy efficient control strategies for the indoor environment when minimizing the use of HVAC systems. These strategies are proposed as alternatives if non-air-conditioned buildings emerge as unsuitable for the proper conservation of exhibits. Indeed, since this approach allows taking into account the storage history and the response of the hygroscopic objects to the historical microclimate, the range of acceptable indoor microclimate conditions for the preservation can be potentially widened. According to this approach, for hygroscopic materials, the target value is the yearly arithmetic mean of the RH measurements; in this case, it is possible to define thresholds higher and lower (±10%) than the average value by admitting more or less wide oscillations to the parameter under consideration before operating the HVAC system or signaling the potential danger due to the variation in progress.

## 3. Results and Discussion

### 3.1. Microclimate Monitoring of Vasari Corridor

This section summarizes the main results of the microclimate monitoring in the Vasari Corridor museum. For each monitoring position, temporal profiles, minimum, medium, and maximum values, daily gradients, and the performance index of the monitored parameters were evaluated. The latter evaluates the quality of the indoor environment in relation to the conservation of the exhibited objects. The monitored data were critically analyzed with the aim of highlighting the main criticalities and potentialities for the artwork conservation in this type of environment.

Figures 8 and 9 present the time profiles of T and RH, respectively, measured in positions A (Lungarno) and B (Ponte Vecchio) for the whole year 2017 with respect to the outdoor values (EXT).

Since the Vasari Corridor is not equipped with an HVAC system (non-air-conditioned museum), the indoor T and RH values monitored inside the museum environments follow the trend outdoor climate conditions. Nevertheless, the indoor conditions are significantly dampened with regard to the hourly and daily fluctuations. The diagrams highlight the stabilization of T and RH inside the Vasari Corridor compared to the outdoor conditions; the maximum outdoor temperature daily variation ($\Delta T_{24}$) is equal to 17.5 °C, and the maximum outdoor relative humidity daily variation ($\Delta RH_{24}$) is equal to 27%, while the indoor daily variation is equal to 2.2 °C for temperature, with a delay of about 4 h, and 5% for relative humidity. The parameters monitored in the two positions (A and B) have a similar trend. However, position B in Ponte Vecchio, characterized by large windows, is much more exposed to the atmospheric agents with respect to position A in Lungarno.

Therefore, it has higher temperatures (and lower relative humidity values) in summer and lower temperatures (and higher relative humidity values) in winter.

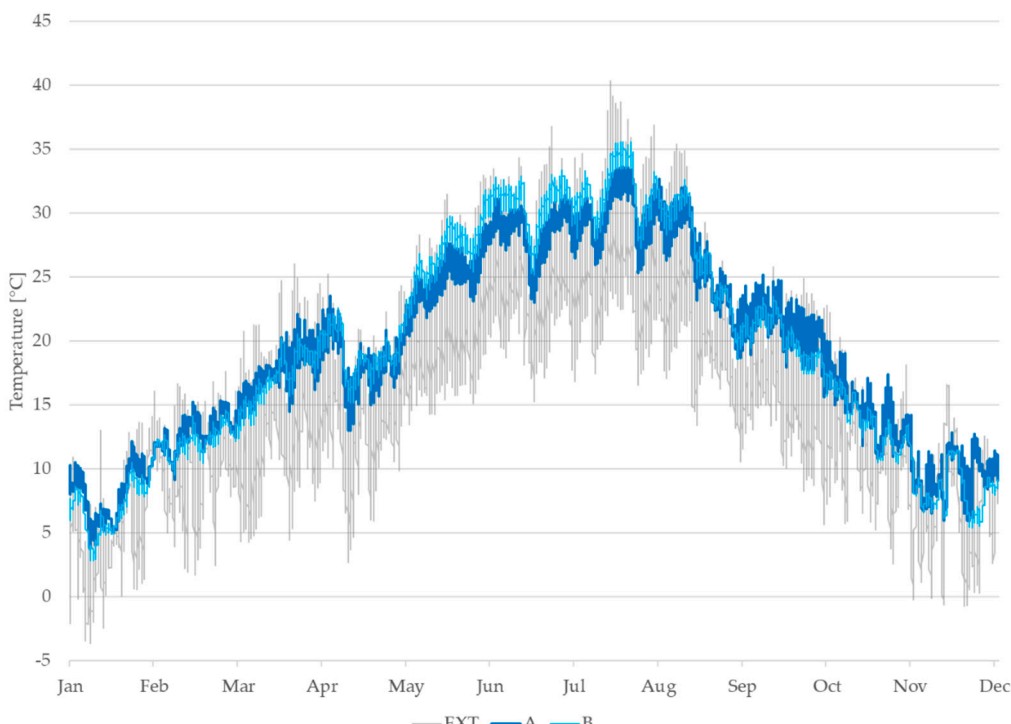

**Figure 8.** Time profiles of T values measured in 2017 in positions A (Lungarno), B (Ponte Vecchio), and outdoor (EXT).

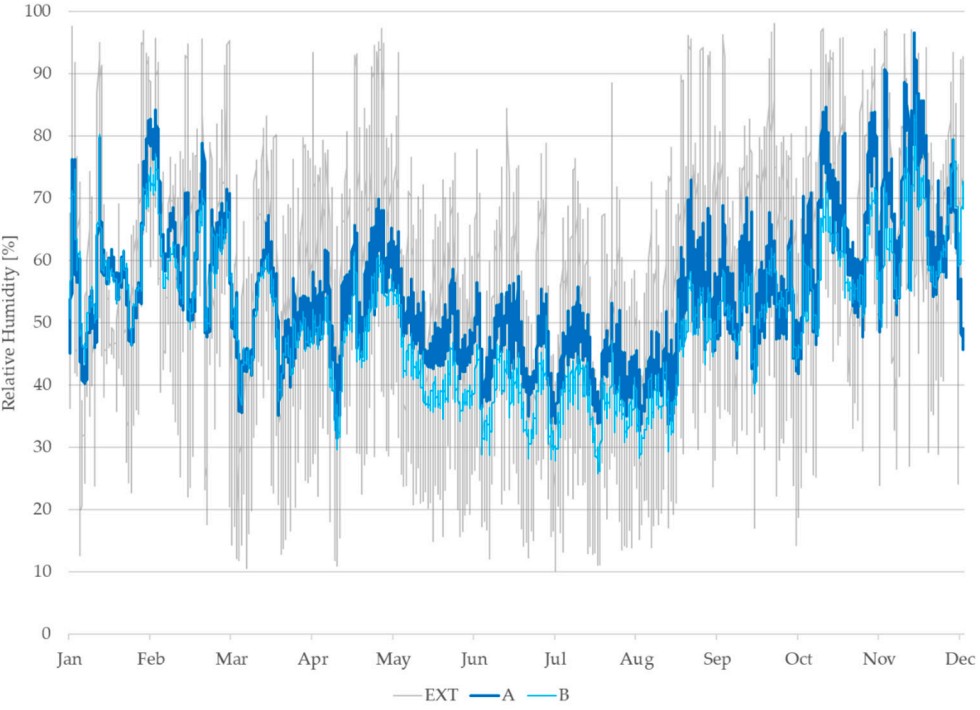

**Figure 9.** Time profiles of RH values measured in 2017 in positions A (Lungarno), B (Ponte Vecchio), and outdoor (EXT).

Figure 10 shows the PI monthly values for T and RH daily variation for the positions A and B. The PI$_{\Delta T24}$ is low, especially during summer period and mostly for position A, while

the RH values remain more constant, and the PI$_{\Delta RH24}$ is higher, especially in position B, throughout the year. This means that the building envelope is not able to dampen the effects of outdoor temperature daily variation within the acceptable range for the conservation of the artworks, especially in the summer period. Instead, the passive control of the daily variation of RH is more effective, especially in position B.

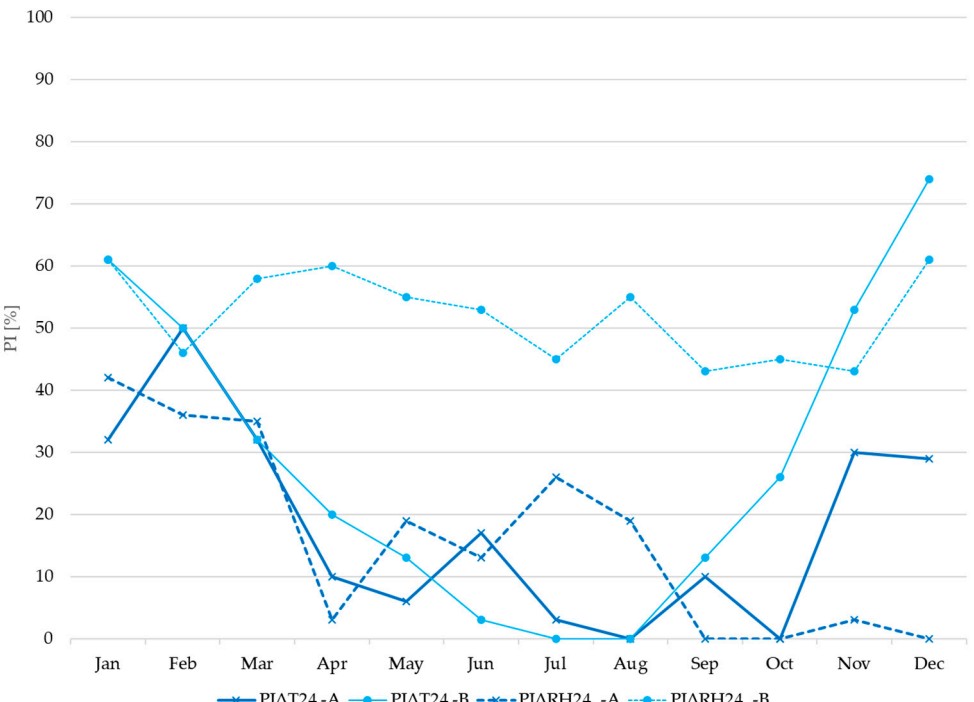

**Figure 10.** PI monthly values of hygrothermal parameters daily variation for position A (Lungarno) and B (Ponte Vecchio) in 2017.

Historical Climate of Vasari Corridor: Calculation of the Target Values for T and RH

A further contribution to the analysis of the measured data is the determination of the historical climate of Vasari Corridor, i.e., the target values included in a statistical range suggested by UNI EN 15757:2010 [17]. In the case of the need of installation of HVAC systems, it is possible to dimension them in order to regulate the microclimatic control as a function of the historical climate typical of each monitored environment, instead of standard reference values.

Figures 11 and 12 show the trend of monthly and annual mean values of T and RH, respectively, for Lungarno (A) and Ponte Vecchio (B). Figures 13–16, instead, depict the range of oscillation of T and RH for position A and B included in the 93rd and seventh percentiles, as suggested by the UNI EN 15757 [17]. In this way, 14% of the largest and riskiest fluctuations are excluded, eliminating equally 7% of the positive and negative peaks of relative humidity that produce excessively damp or dry conditions.

Therefore, by taking into account the historical climate, the HVAC systems should be operated only when the T or RH values exceed the aforementioned range, which are very limited, thus limiting energy consumption to the minimum required so as not to compromise, but rather to favor, the preservation of artworks. Further similar insights can be made by working on a monthly statistical basis instead of annual or seasonal.

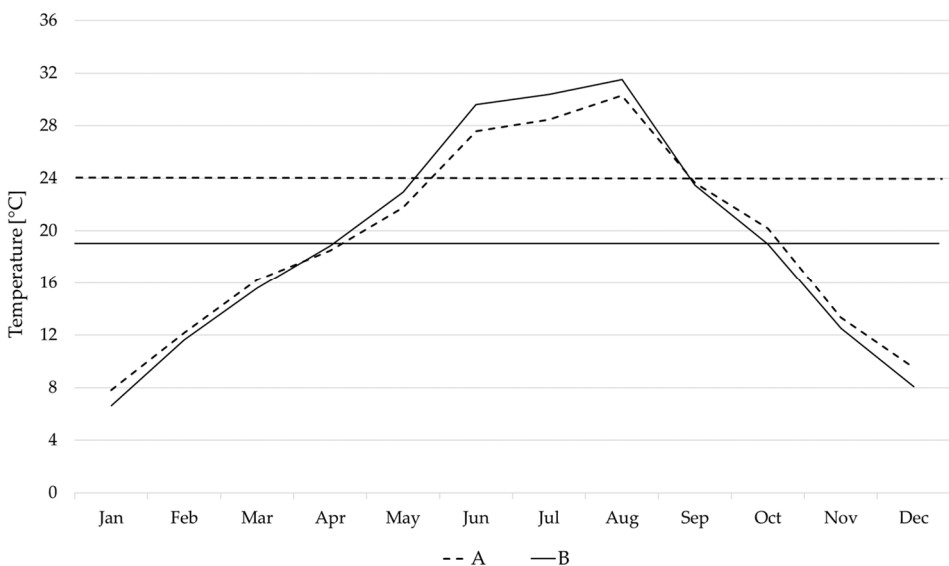

**Figure 11.** Trend of monthly and annual (horizontal line) mean values of T for Lungarno (**A**) and Ponte Vecchio (**B**).

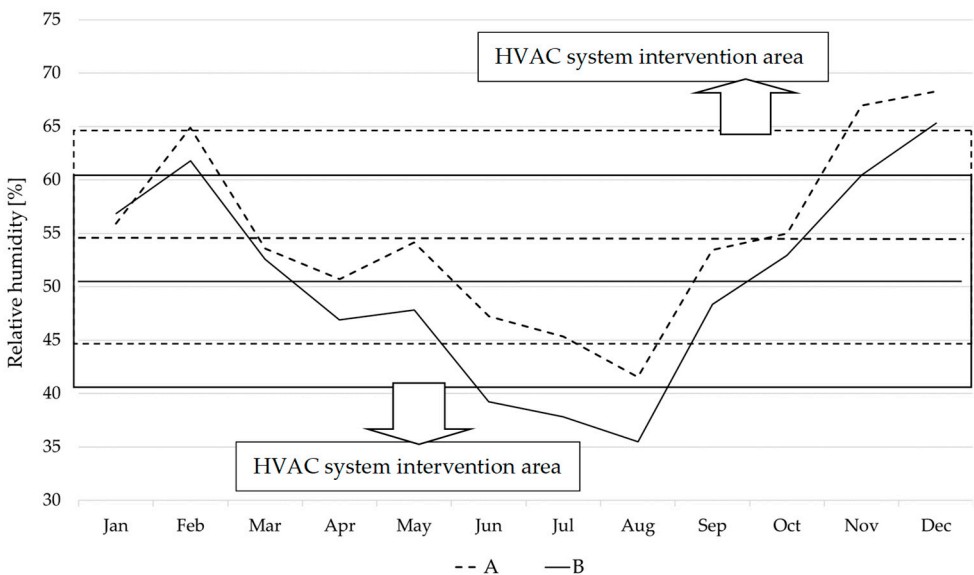

**Figure 12.** Trend of monthly and annual (horizontal line) mean values of RH for Lungarno (**A**) and Ponte Vecchio (**B**). The rectangles define the range of acceptable values for the two zones.

Indeed, if we compare the acceptable T and RH range derived from the analysis of historical climate with the recommended reference values for the conservation of paintings, reported in the previous Table 5, a significant difference emerges, with much wider margins of tolerance of variations. Given that the decision on the harmlessness or otherwise of the existing climatic conditions must be left to the museum conservator, this approach usually allows objective temperatures and relative humidity ranges more flexible than the individual target values that are commonly accepted as ideal conditions for the conservation of cultural heritage. A comparison of the results is analyzed in detail in Section 3.3.

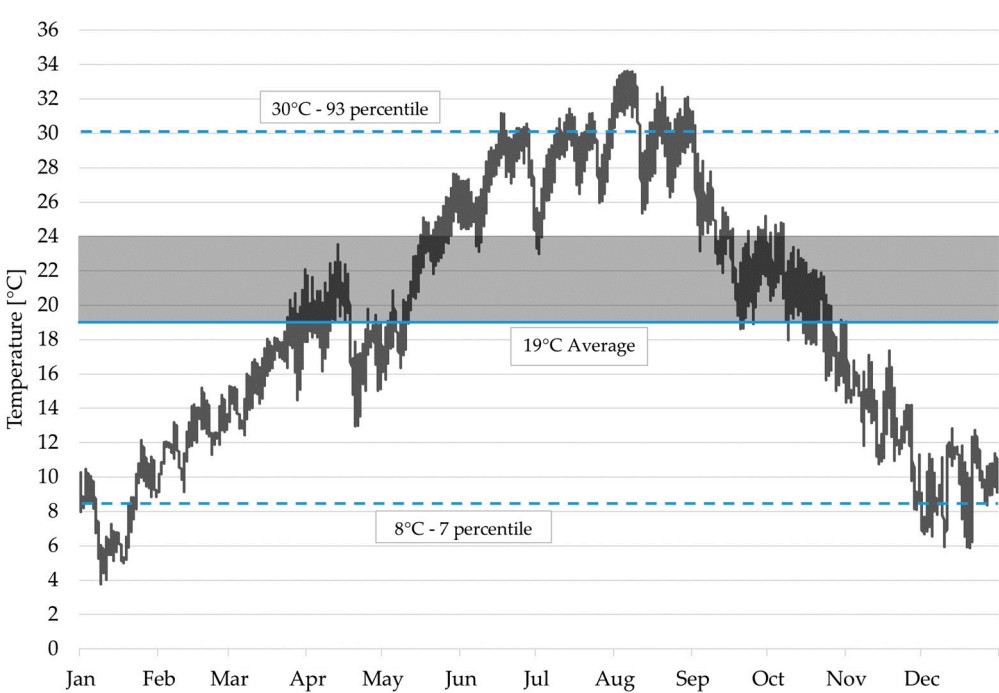

**Figure 13.** Trend of historic climate hourly values of T and statistical limits of 93rd and seventh percentiles in position A vs. the range recommended by the Italian D.M. 10.05.2001 [16] (gray band).

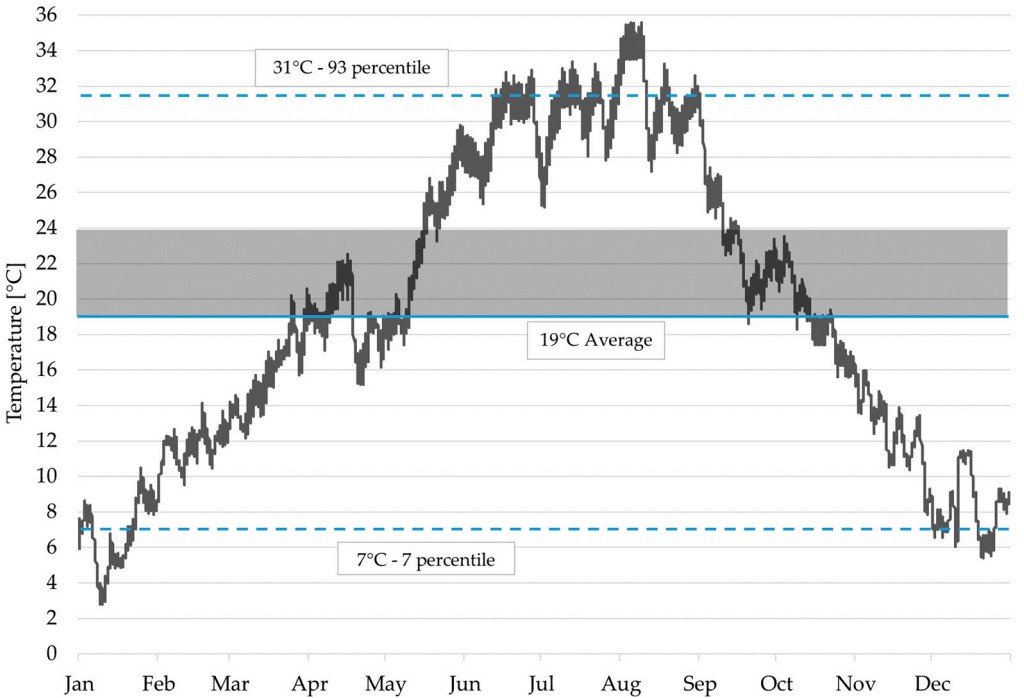

**Figure 14.** Trend of historic climate hourly values of T and statistical limits of 93rd and seventh percentiles in position B vs. the range recommended by the Italian D.M. 10.05.2001 [16] (gray band).

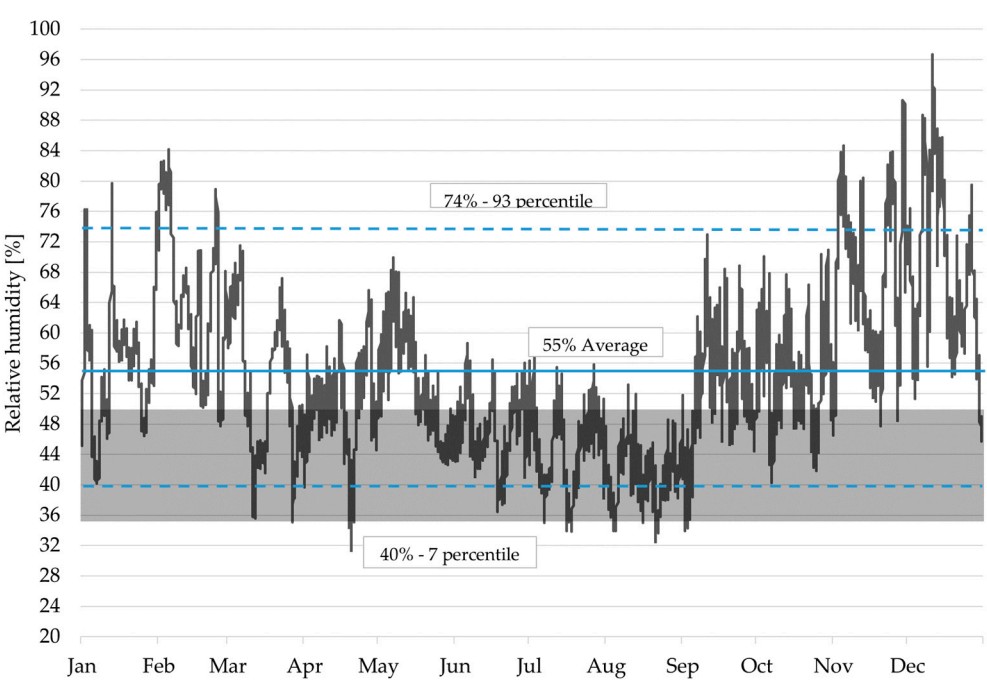

**Figure 15.** Trend of historic climate hourly values of RH and statistical limits of 93rd and seventh percentiles in position A vs. the range recommended by the Italian D.M. 10.05.2001 [16] (gray band).

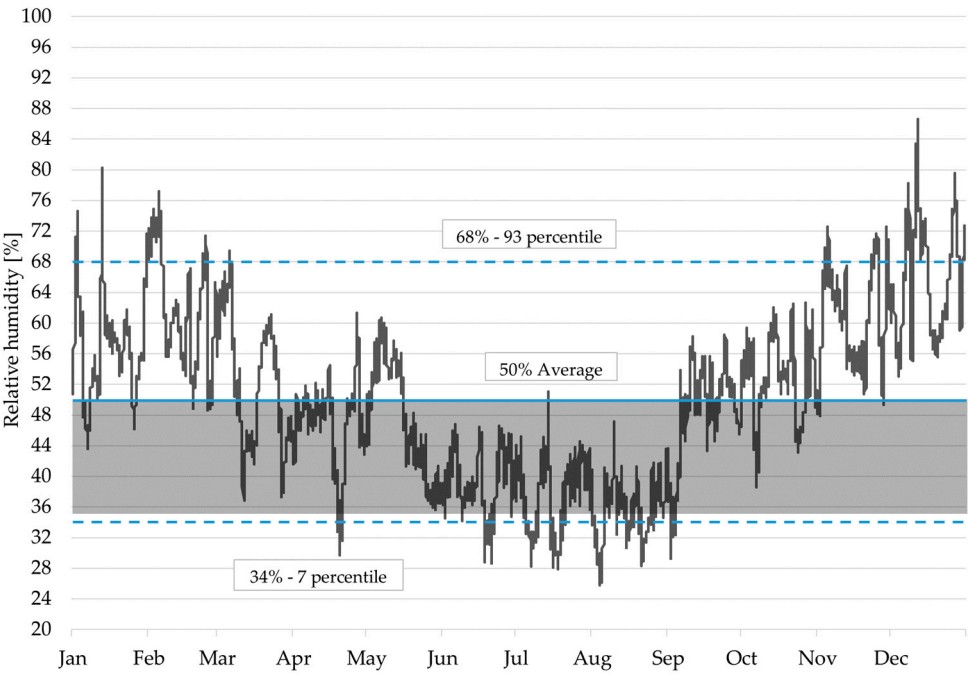

**Figure 16.** Trend of historic climate hourly values of RH and statistical limits of 93rd and seventh percentiles in position B vs. the range recommended by the Italian D.M. 10.05.2001 [16] (gray band).

### 3.2. Microclimate Monitoring of La Specola Museum

This section summarizes the main results of the microclimate monitoring in the La Specola museum. For each analyzed zone (rooms X and XXII), temporal profiles, minimum, medium, and maximum values, daily gradients, and the performance index of the monitored parameters are analyzed. Figures 17 and 18 show the time profiles of T and RH in the two monitored positions for the whole analyzed year (May 2012–April 2013) with respect to the corresponding outdoor values (EXT).

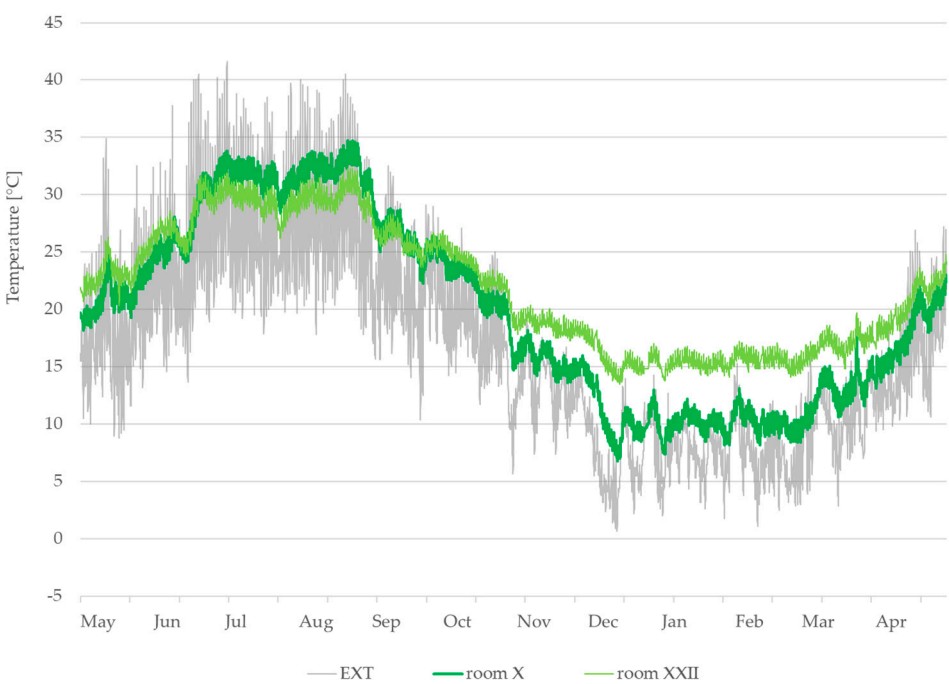

**Figure 17.** Time profiles of temperature T values measured from May 2012 to April 2013 in rooms X and XXII, and outdoors (EXT).

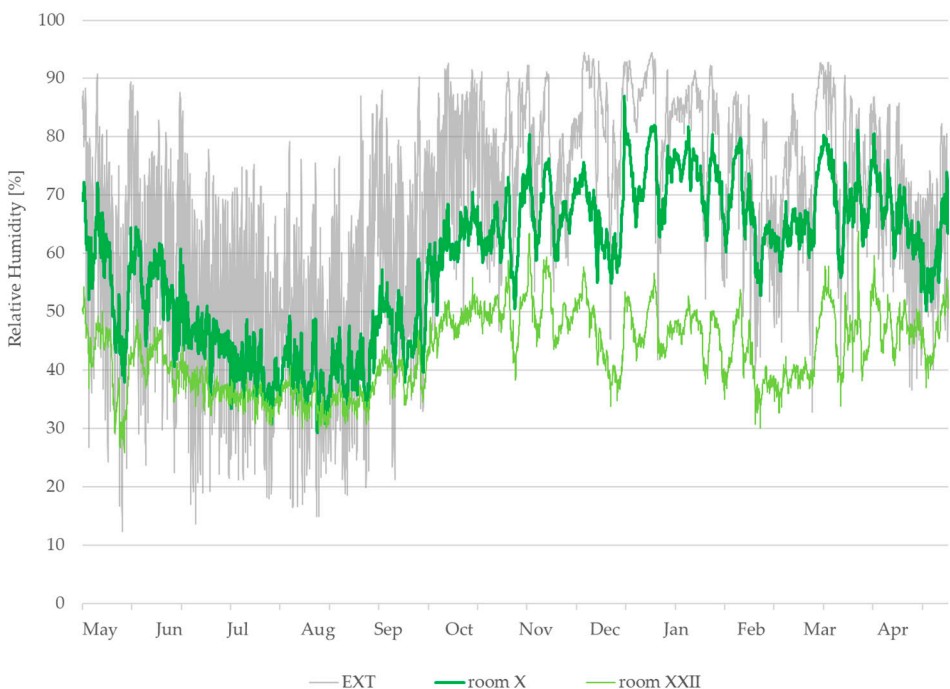

**Figure 18.** Time profiles of relative humidity RH values measured from May 2012 to April 2013 in rooms X and XXII, and outdoors (EXT).

The T and RH trends were similar for the two analyzed rooms (X and XXII) and follow the outdoor climate, especially in room X. On the contrary, in room XXII the indoor microclimate was significantly dampened in terms of both temperature and relative humidity compared to the outdoor fluctuations; this was due to the fact that the room is an internal room without external walls and, thus, more protected from the outdoors.

Figure 19 shows the PI monthly values for T and RH daily variation for rooms X and XXII. The $PI_{\Delta T24}$ is low, especially during summer period, and the values are similar for the two rooms, while the $PI_{\Delta RH24}$ is higher, especially in room XXII, throughout the year.

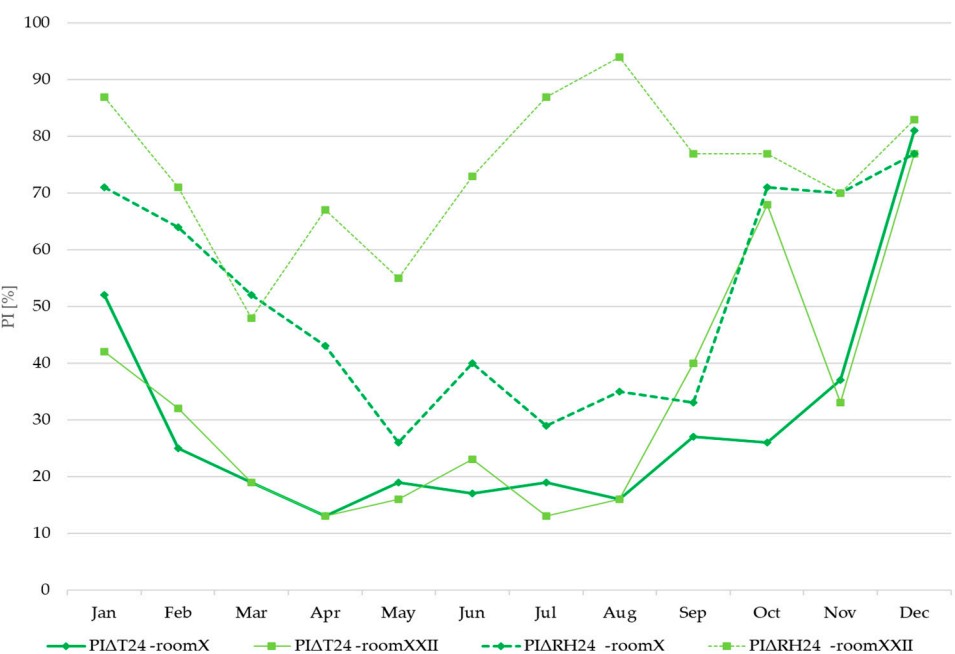

**Figure 19.** PI monthly values of hygrothermal parameters daily variation for room X and XXII.

Historical Climate of La Specola: Calculation of the Target Values for T and RH

For La Specola, the values of the conservation range in terms of T and RH were also evaluated according to the assessment of the historical climate.

Figures 20 and 21 show the trend of monthly and annual mean values of T and RH, respectively, for rooms X and XXII. In room XXII, the RH monthly fluctuation was always within the range of acceptability.

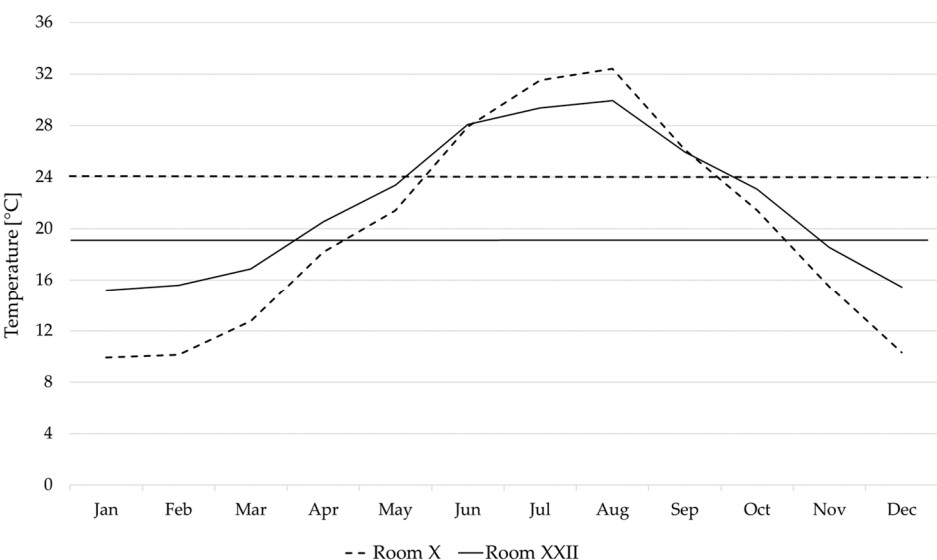

**Figure 20.** Trend of monthly and annual (horizontal line) mean values of T for rooms X and XXII.

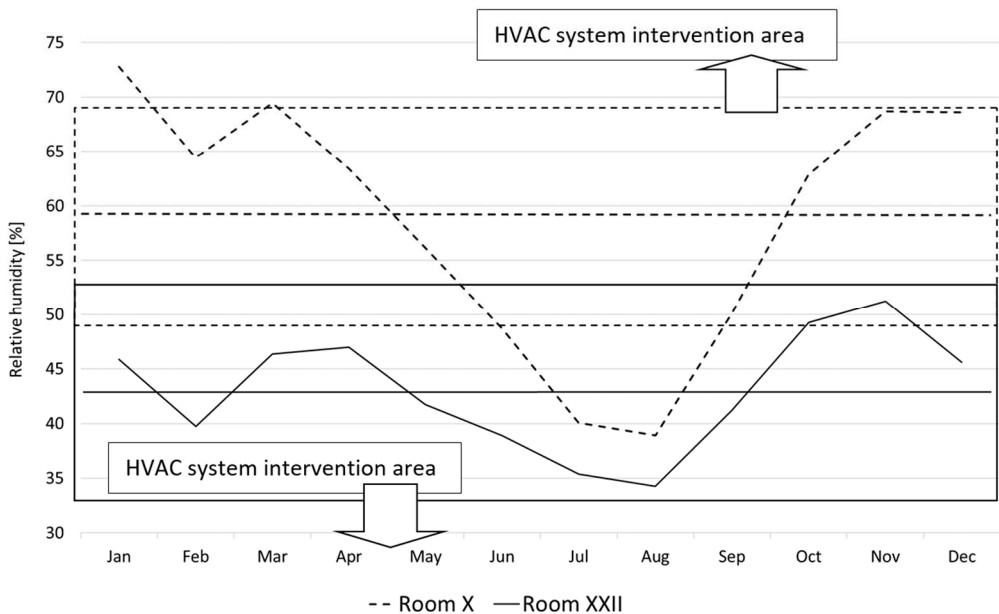

**Figure 21.** Trend of monthly and annual (horizontal line) mean values of RH for rooms X and XXII. The rectangles define the range of acceptable values for the two zones.

The ranges of oscillation of T and RH included in the 93rd and seventh percentile, as suggested by the UNI EN 15757 [17], for rooms X and XXII are shown in Figures 22–25.

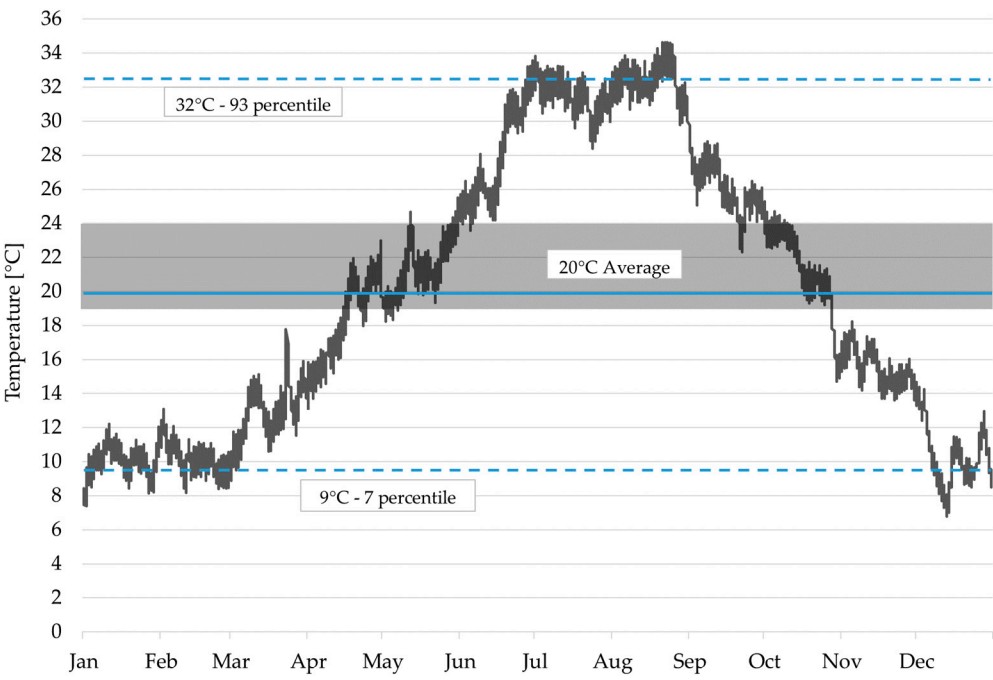

**Figure 22.** Trend of historic climate hourly values of T and statistical limits of 93rd and seventh percentiles in room X vs. the range recommended by the Italian D.M. 10.05.2001 [16] (gray band).

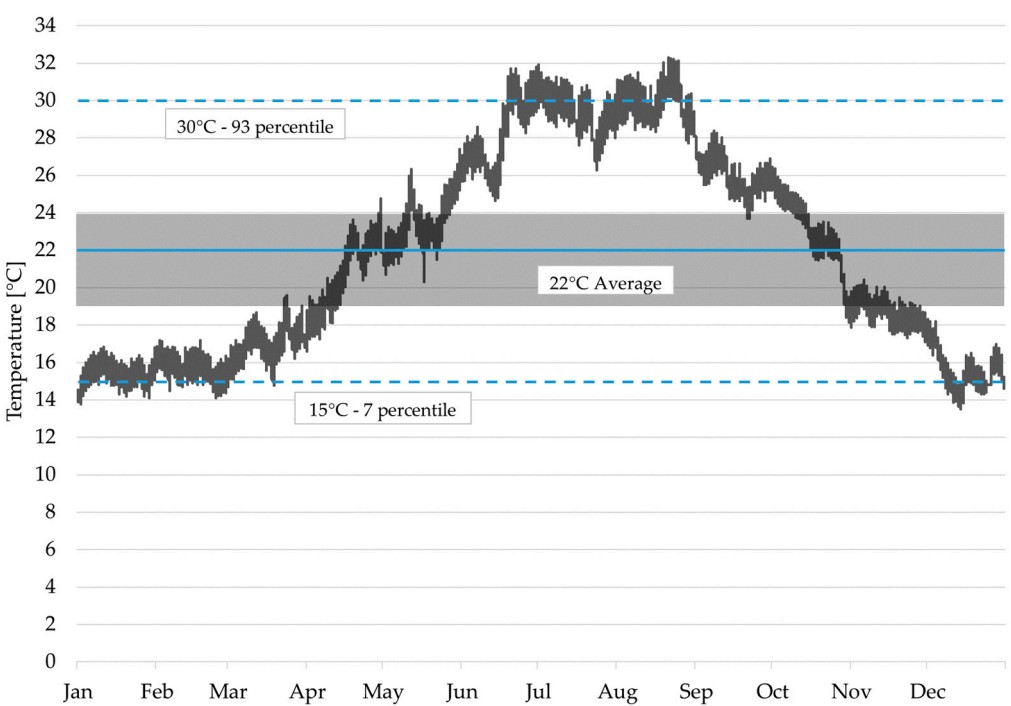

**Figure 23.** Trend of historic climate hourly values of T and statistical limits of 93rd and seventh percentiles in room XXII vs. the range recommended by the Italian D.M. 10.05.2001 [16] (gray band).

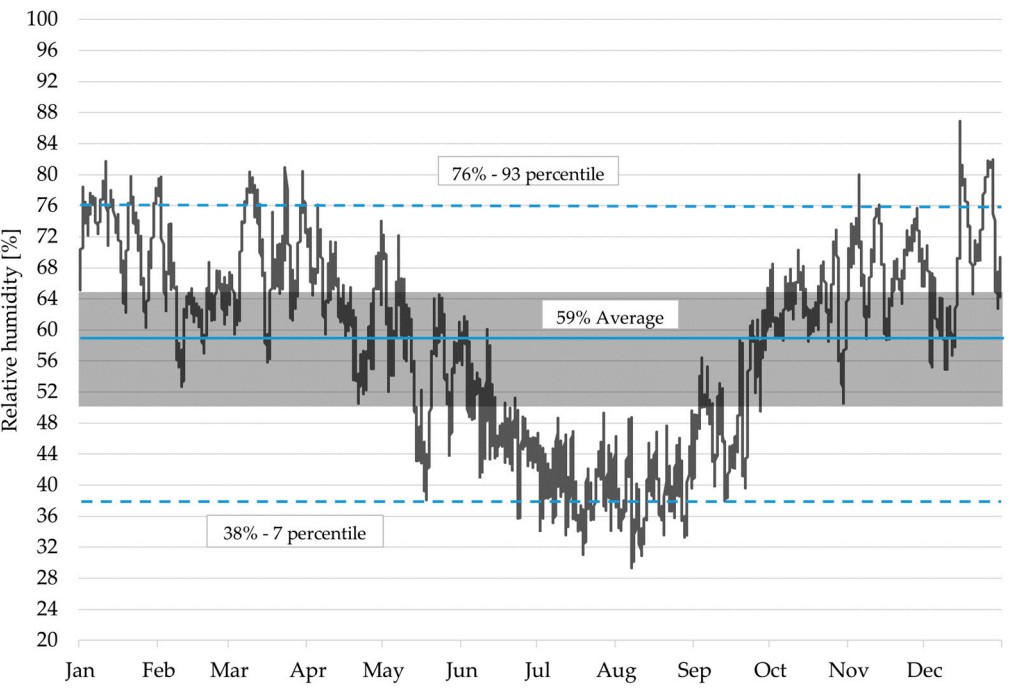

**Figure 24.** Trend of historic climate hourly values of RH and statistical limits of 93rd and seventh percentiles in room X vs. the range recommended by the Italian D.M. 10.05.2001 [16] (gray band).

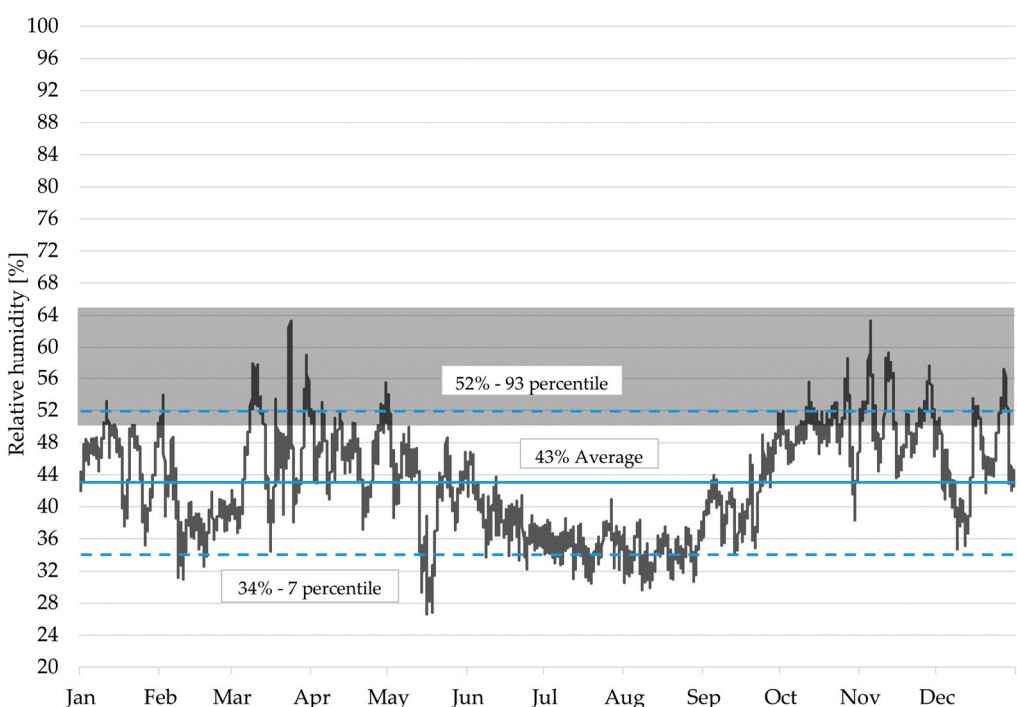

**Figure 25.** Trend of historic climate hourly values of RH and statistical limits of 93rd and seventh percentiles in room XXII vs. the range recommended by the Italian D.M. 10.05.2001 [16] (gray band).

In line with the results obtained for the Vasari Corridor, if the range of statistical values deriving from the analysis of the historical climate in La Specola museum is compared with the recommended reference values for the conservation according to the standards reported in Table 5, a substantial difference emerges with much wider margins of tolerance. It is also noted that, since the trend of RH in room XXII is flatter, the percentile limits are close to the range that is obtained from the annual average with limits (±10%). A comparison of the results is analyzed in detail in Section 3.3.

### 3.3. Comparison between Vasari Corridor and La Specola

Figures 26 and 27 compare the average monthly values of indoor temperature and relative humidity, respectively, monitored in Vasari Corridor and La Specola. The results clearly show that La Specola room X, characterized by low thermal inertia and with two windows exposed to solar radiation, behaves similarly to the zones of Vasari Corridor, in terms of both temperature and relative humidity. On the contrary, room XXII, characterized by higher thermal inertia and located inside the building without external walls and, thus, more protected from the outdoors, presents a flatter trend of indoor temperatures along the year, up to about 8 °C higher in colder months and 3 °C lower in hotter months, and significantly lower values of relative humidity throughout the year, up to about 26% difference.

Table 6 compares the values of the statistical parameters characterizing the historical climate of temperature and relative humidity in each analyzed zone compared with the reference value coming from D.M. 10.05.2001 on the annual basis. In all four zones, the range of preservation coming from the analysis of the historical climate, to which the objects are now acclimatized, is much wider and rather different than that suggested by the D.M. standards.

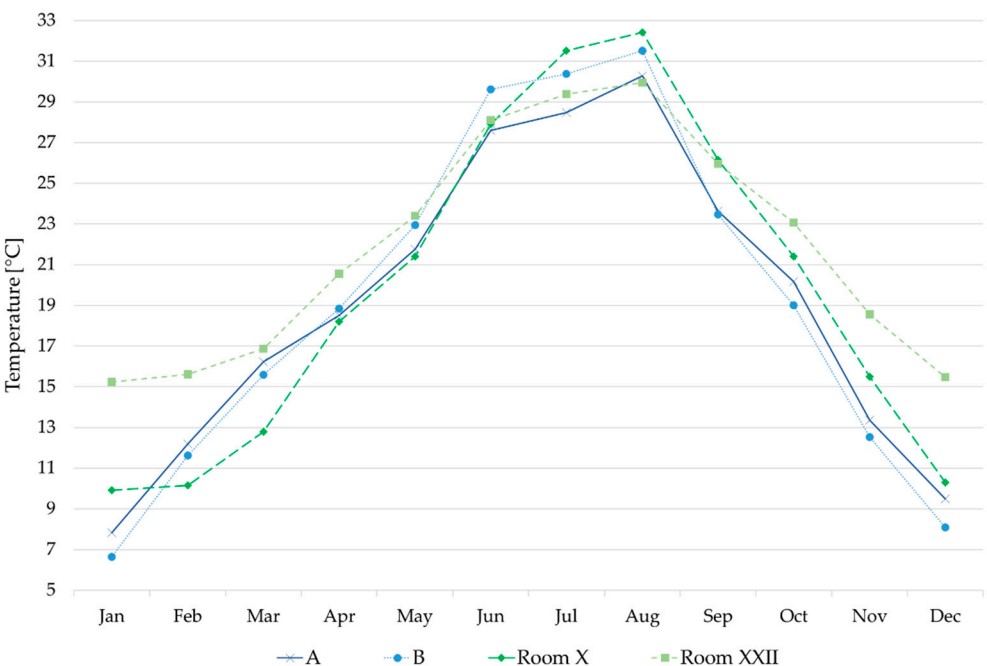

**Figure 26.** Trend of average indoor monthly temperature (T) measured in the analyzed zones of Vasari Corridor and La Specola museum.

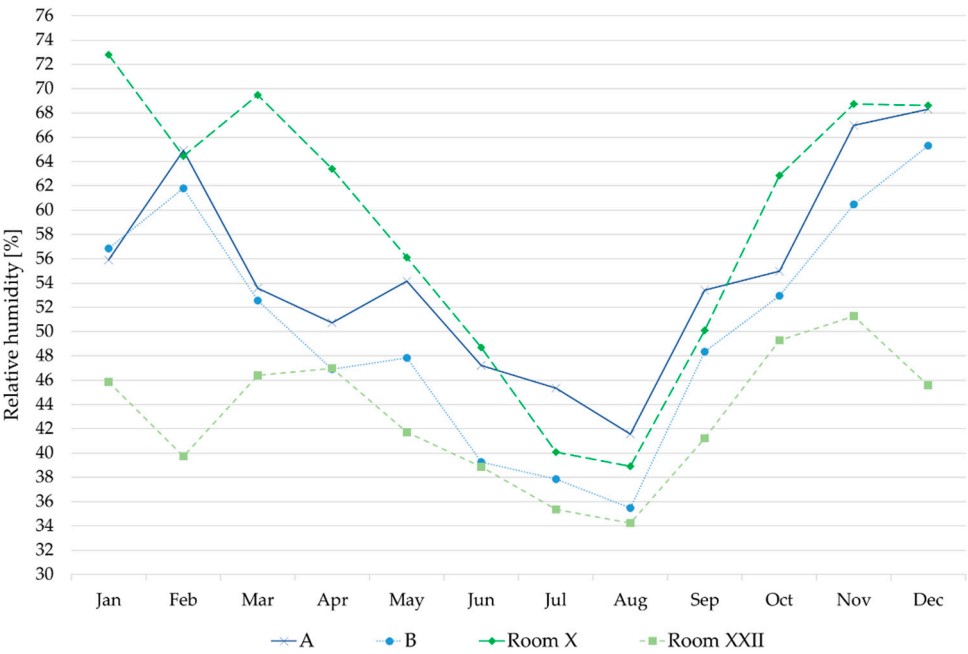

**Figure 27.** Trend of average indoor monthly relative humidity (RH) measured in the analyzed zones of Vasari Corridor and La Specola museum.

**Table 6.** Values of average T and RH by historical climate.

| Parameter | Point of Monitoring | Average Yearly Value | Percentile | | Mean ± 10% * | | Reference Value (D.M. 10.05.2001) |
|---|---|---|---|---|---|---|---|
| | | | 7 | 93 | −10 | +10 | |
| T [°C] | Lungarno (A) | 19.2 | 8 | 30 | - | - | 19 ÷ 24 |
| | Ponte Vecchio (B) | 19.2 | 7 | 31 | - | - | |
| | Room X | 19.9 | 9 | 32 | - | - | 19 ÷ 24 |
| | Room XXII | 21.9 | 15 | 30 | - | - | |
| RH [%] | Lungarno (A) | 54.7 | 40 | 74 | 45 | 65 | 35 ÷ 50 |
| | Ponte Vecchio (B) | 50.4 | 34 | 68 | 40 | 60 | |
| | Room X | 58.6 | 38 | 76 | 49 | 69 | 50 ÷ 65 |
| | Room XXII | 43.1 | 34 | 52 | 33 | 53 | |

* This range is foreseen by the standard UNI EN 15757:2010 [17] only for relative humidity.

Table 7 summarizes the maximum daily gradients of T ($\Delta T_{24max}$) and RH ($\Delta RH_{24max}$) together with the PI of the hygrothermal parameters (T, RH, $\Delta T_{24}$, and $\Delta RH_{24}$) for the four analyzed zones in the two buildings, i.e., Lungarno (point A) and Ponte Vecchio (point B) in Vasari Corridor and room X and room XXII in La Specola. The PI yearly values calculated accordingly to the reference values recommended by the Italian standard D.M. 10.05.2001 (third column) were compared with those calculated on the basis of the historical climate (fourth column). The comparison shows that Vasari Corridor zones present much higher peak daily gradient, in terms of both temperature and relative humidity, than La Specola rooms. Accordingly, on average, the percentage of time when the daily gradients of temperature and relative humidity are within the range of acceptability is higher in La Specola rooms than in Vasari Corridor. In particular, the zone with higher $PI_{\Delta T24}$ (about 33%) and better $PI_{\Delta RH24}$ (about 74%) is room XXII, as expected from previous results.

**Table 7.** Maximum yearly gradient and PI values of hygrothermal parameters in the four analyzed zones: A and B in Vasari Corridor and rooms X and XXII in La Specola museum.

| Parameter | Point of Monitoring | Yearly Value (D.M. 10.05.2001) | Yearly Value (Historical Climate) |
|---|---|---|---|
| $\Delta T_{24\,max}$ [°C] | A | 5.1 | 5.1 |
| | B | 4.5 | 4.5 |
| | X | 2.3 | 2.3 |
| | XXII | 2.8 | 2.8 |
| $\Delta RH_{24\,max}$ [%] | A | 32.0 | 32.0 |
| | B | 23.0 | 23.0 |
| | X | 9.1 | 9.1 |
| | XXII | 7.5 | 7.5 |
| $PI_{\Delta T24}$ [%] | A | 18.0 | 18.0 |
| | B | 29.0 | 29.0 |
| | X | 29.0 | 29.0 |
| | XXII | 33.0 | 33.0 |
| $PI_{\Delta RH24}$ [%] | A | 16.0 | 16.0 |
| | B | 52.0 | 52.0 |
| | X | 51.0 | 51.0 |
| | XXII | 74.0 | 74.0 |
| $PI_T$ [%] | A | 19.0 | 86.0 |
| | B | 15.0 | 84.0 |
| | X | 19.0 | 87.0 |
| | XXII | 19.0 | 84.0 |
| $PI_{RH}$ [%] | A | 37.0 | 86.0 |
| | B | 39.0 | 87.0 |
| | X | 36.0 | 88.0 |
| | XXII | 16.0 | 86.0 |

Considering the reference values recommended by the Italian standard D.M. 10.05.2001, for both the museums, the analysis of indoor microclimate parameters demonstrates that

T and RH conditions were not reasonably acceptable for the preservation of the kind of artworks exhibited during the majority of the monitored period. In spring and autumn, the indoor T and RH are within the recommended range, while, in the extreme conditions of summer and winter, the temperature is above or below the limits, respectively. As expected, due to monitored indoor T and RH values outside the recommended range, the yearly $PI_T$ ranges from 15% (position B) to 19% (position A, room X and XXII), and the yearly $PI_{RH}$ ranges from 16% (room XXII) to 39% (position B). In particular, for room XXII, this $PI_{RH}$ low value is due to the low RH values characteristic of the room, which is less affected by the external climate.

On the contrary, considering the reference values calculated on the basis of the historical climate, according to UNI EN 15757 [17], for both museums, the yearly PI values are quite high, with $PI_T$ ranging from 84% (position B and room XXII) to 87% (room X), and $PI_{RH}$ ranging from 86% (position A and room XXII) to 88% (room X).

To assess the thermal energy behavior of buildings with respect to the climatic stresses and to compare the performance of the two analyzed museum buildings, the difference between indoor and outdoor values of temperature and relative humidity was also evaluated. Figure 28 shows the trend of average monthly temperature differences for the Vasari Corridor and La Specola.

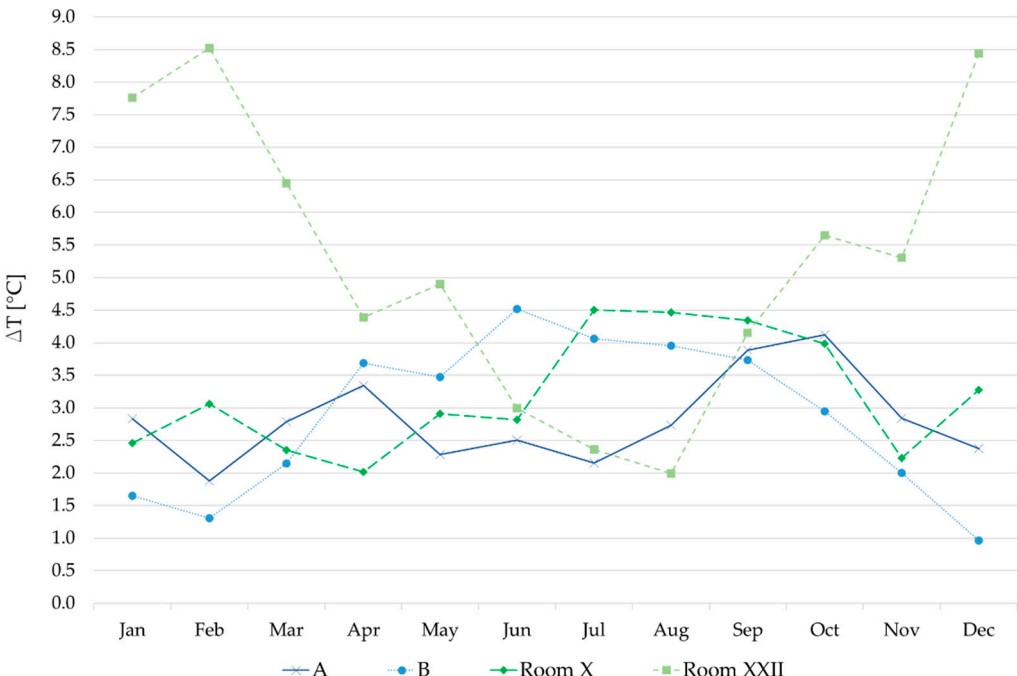

**Figure 28.** Trend of monthly temperature difference between indoor and outdoor for the analyzed zones of Vasari Corridor and La Specola museum.

For La Specola, the following can be observed:
- Room XXII (internal room, no external walls, with skylight): the average temperature difference is equal to 5.2 °C and decreases in the warm season (from April to September), while it shows peak values in the coldest months (January, February, and December). Indeed, the innermost position allows a notable buffering from the external cold climate, but provides less protection from summer radiation coming from the skylight and low chances of air exchange in the hottest months. The lowest values of temperature difference occur in late spring/mid-autumn, demonstrating that the effects of solar radiation coming from the unshielded skylight significantly influence the internal temperature.
- Room X (corner room with two external walls): the average temperature difference is equal to 3.2 °C and has almost flat trend from November to June, with values below the

average, when the indoor temperature follows more closely the trend of the external temperature. On the contrary, the difference tends to increase and remain almost constant in the hottest months from July to October, when the room is shaded from the outdoor conditions (mainly solar radiation) even more than room XXII, because of the shielded windows.

The effects of incoming solar radiation are also relevant in the Vasari Corridor. In particular, the following can be observed:

- Lungarno—position A (small windows and southern orientation): the average temperature difference is equal to 2.8 °C and the trend is quite flat, probably due to the protection of the nearby buildings on the north side, except for a slight increase in September and October.
- Ponte Vecchio—position B (large windows and east–west orientation): the average temperature difference is equal to 2.9 °C, but its trend is in contrast with the previous zone. Indeed, starting from April, it increases until September and then decreases, reaching the lowest values, compared to all the zones in both buildings, in the coldest months (January, February, and December) to less than 1 °C.

Overall, the discrepancy in the trend of the temperature difference occurs between May and August, i.e., in the hottest months (end of spring–summer), with Lungarno being more exposed to the dominant solar radiation (south orientation).

Lastly, in general, it can be noted that, with the exception of room XXII in La Specola museum, in the other analyzed zones, the difference in average monthly temperatures is maintained around 3 °C. Accordingly, the passive performance of the building envelope is lower than what is required to ensure the indoor microclimate conditions able to preserve the exhibits without the operation of a HVAC system. Therefore, the improvement of the energy performance of the actual building envelope, e.g., higher thermal insulation and solar shading, could enhance the hygro-thermal behavior of the building. However, historical buildings, such as those analyzed in this study, have strict architectural and preservation constraints, which often limit the feasibility of sensitive physical retrofits [43]. This study proposes an alternative strategy to minimize the energy consumed for the operation of the HVAC system, while ensuring the preservation of the exhibits; it is proposed to use the hygrothermal parameters (air temperature and relative humidity) that characterize the historical climate in each museum zone to control the operation of HVAC system in the zone. The historical climate, indeed, provides more flexible ranges than the reference values according to standards for the conservation of cultural heritage [15,16], thus reducing the energy needs for the environmental control.

## 4. Conclusions

In light of the current energy crisis and the need to improve building energy efficiency, this study analyzed the suitability of non-air-conditioned historical building museums to properly preserve different types of artworks. To this aim, two museums in Florence, Italy, namely, the Vasari Corridor and La Specola Museum were investigated as case studies. These museums host the exhibitions of artworks and objects that require specific microclimate for optimal preservation, according to the reference technical standards.

The existing thermophysical survey and 1 year microclimate monitoring data provided the necessary information to understand the behavior of each building in terms of the trend of air temperature and relative humidity in the absence of an air-conditioning system.

The analysis of data monitored during the year 2013 for La Specola and 2017 for the Vasari Corridor showed that the temperature and relative humidity trends inside the uncontrolled environments follow the outdoor climate, albeit markedly dampened in sudden hourly and daily changes. In particular, zones characterized by envelope with low inertia and/or higher exposure to external conditions, such as room X in La Specola and Ponte Vecchio in the Vasari Corridor, have indoor temperature trends similar to the external temperature trend. Moreover, the low thermal inertia of the Vasari Corridor, with large glazed surfaces and lightweight external walls and roof, is the cause of rapid variations

in temperature (over 2 °C) and relative humidity (up to 5%), even within the same day. However, significant differences were detected between Lungarno and Ponte Vecchio, which have different exposure to solar radiation and ratio of transparent surfaces. On the one hand, position B in Ponte Vecchio, characterized by large windows, is much more exposed to external conditions, which results in higher temperature (and lower relative humidity) in summer and vice versa in winter. On the other hand, room XXII in La Specola, which presents higher thermal inertia, and which is located inside the building and, thus, more protected from the outdoors, presents a flatter trend of indoor temperatures and significantly lower values of relative humidity throughout the year.

Nevertheless, in all the analyzed non-air-conditioned zones of the two museums, the indoor temperature and relative humidity values are outside the recommended range for the preservation of the kind of objects exhibited according to the reference technical standard for most of the year. This result suggests the need to install an air-conditioning system for the proper conservation of the artwork.

However, give the current energy crisis, which could even prevent the opening of smaller museums due to unattainable operation costs, to minimize the energy consumption, while preserving the artworks, this work suggests the use of a different, more variation-tolerant approach, according to the standard UNI EN 1575:2010. This approach is based on the analysis of the historical climate within the exhibition zones. The campaign of indoor microclimate monitoring conducted over a period of at least 1 year allows defining the historical climate in each exhibition zone. The analysis of the measured data using this approach allows wider objective temperature and relative humidity ranges than the individual target values that are commonly accepted as ideal conditions for the conservation of cultural heritage by the reference technical standard. Indeed, when the artworks are observed to have suffered no damage, the historical climate can be used as the actual reference range for the proper preservation of the artworks. The results of this work show that the switch to these wider ranges could limit the energy consumption for the required air conditioning without compromising the preservation of the exhibits. Although these outcomes were observed for two specific case study historical museum buildings, the same approach can be similarly implemented in the numerous museums that have similar characteristics in terms of constructive typology and climate boundary conditions.

**Author Contributions:** Conceptualization, F.S., C.C. and G.C.; methodology, F.S., C.C., G.C. and C.P.; supervision, F.S., C.C., G.C. and C.P.; writing—original draft, F.S., G.C. and C.P.; writing—review and editing, F.S., C.C., G.C. and C.P. All authors have read and agreed to the published version of the manuscript.

**Funding:** This research received no external funding.

**Institutional Review Board Statement:** Not applicable.

**Acknowledgments:** The authors acknowledge Eike Schmidt and Giuseppe Russo of Gallerie degli Uffizi for their collaboration and availability. Moreover, the authors would like to thank Emilio Borchi of the Fondazione Osservatorio Ximeniano Onlus and Alessandro Zandei of IBIMET—Institute of Biometeorology, National Research Council, for providing the weather data. Acknowledgements are also due to Marco Benvenuti, Angela Di Ciommo, Fausto Barbagli, Gianna Innocenti, and Claudia Corti of the Natural History Museum of Florence for providing co-operation during the process of collecting data. C.P. would like to thank the Italian funding programme Fondo Sociale Europeo REACT EU—Programma Operativo Nazionale Ricerca e Innovazione 2014–2020 (European Social Fund REACT EU—National Operational Program for Research and Innovation 2014–2020) (D.M. n.1062 of 10 August 2021) for supporting her research.

**Conflicts of Interest:** The authors declare no conflict of interest.

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
