# Peer review of "Assessment of the Suitability of Non-Air-Conditioned Historical Buildings for Artwork Conservation: Comparing the Microclimate Monitoring in Vasari Corridor and La Specola Museum in Florence"

_applsci, doi:10.3390/app122211632_

Round 1
Reviewer 1 Report
The paper covers the comparison of static preservation hygrothermal parameters compared to those defined using historical climate data for four free-running (non-air-conditioned) spaces in two linked buildings in Florence. Whilst similar studies have been undertaken for air-conditioned buildings (these are however not noted by the authors and should be) there has been less undertaken for free-running exhibition spaces. As such the paper has value.
More could be done to place the research in context. This is not the first time that fixed conservation/preservation hygrothermal parameters has been proposed and this should be acknowledged. Equally there is no mention of the potential for improvements to building envelope performance. Whilst it is understood that this is not the focus of the paper, it is strange to have no mention of this. Parts of the buildings in question were reconstructed in the 20th century and sensitive building fabric improvements should not be out of the question.
There are some errors in English throughout the article, and some areas requiring further clarification. Please see attached detailed mark-up.

Author Response
Response to Reviewer 1 Comments
General comment: The paper covers the comparison of static preservation hygrothermal parameters compared to those defined using historical climate data for four free-running (non-air-conditioned) spaces in two linked buildings in Florence. Whilst similar studies have been undertaken for air-conditioned buildings (these are however not noted by the authors and should be) there has been less undertaken for free-running exhibition spaces. As such the paper has value.
More could be done to place the research in context. This is not the first time that fixed conservation/preservation hygrothermal parameters has been proposed and this should be acknowledged.
Equally there is no mention of the potential for improvements to building envelope performance. Whilst it is understood that this is not the focus of the paper, it is strange to have no mention of this. Parts of the buildings in question were reconstructed in the 20th century and sensitive building fabric improvements should not be out of the question.
There are some errors in English throughout the article, and some areas requiring further clarification. Please see attached detailed mark-up.
Response: The authors thank the reviewer very much for the positive feedback and for the comments; they are valuable and very helpful for revising and improving the paper. Based on this review, careful modifications have been made to the manuscript, including English revision. All changes are tracked in the revised version of the manuscript. The detailed responses to all reviewer’s comments are reported in the attached file.

Reviewer 2 Report
1.
L. 678 “This result suggests the need of installing an air-conditioning system for the proper conservation of the artworks” but energy costs. Hence, the authors proposed to apply a different, more variation-tolerant approach, according to the standard EN 1575:2010, based “on the analysis of the historical climate within the exhibition zones.”
2.
Indeed, nothing new. For some examples, the fourth one is written by the same authors.
1) Camuffo, D. Microclimate for cultural heritage; Elsevier: 1998.
2) Loupa, G.; Charpantidou, E.; Kioutsioukis, I.; Rapsomanikis, S. Indoor microclimate, ozone and nitrogen oxides in two medieval churches in Cyprus. Atmospheric Environment 2006, 40, 7457-7466, doi:10.1016/j.atmosenv.2006.07.015.
3) Corgnati, S.P.; Fabi, V.; Filippi, M. A methodology for microclimatic quality evaluation in museums: Application to a temporary exhibit. Building and Environment 2009, 44, 1253-1260, doi:https://doi.org/10.1016/j.buildenv.2008.09.012.
4) Sciurpi, F.; Carletti, C.; Cellai, G.; Pierangioli, L. Environmental monitoring and microclimatic control strategies in “La Specola” museum of Florence. Energy and Buildings 2015, 95, 190-201, doi:https://doi.org/10.1016/j.enbuild.2014.10.061.
5) Schito, E.; Testi, D.; Grassi, W. A Proposal for New Microclimate Indexes for the Evaluation of Indoor Air Quality in Museums. Buildings 2016, 6, 41.
3.
Also, L. 366-367 “This index allows the assessment of the quality of the indoor environment in relation to the conservation capability of the object exhibited inside the zone”.
This in not IAQ, is only microclimatic conditions. Aerosol is a major concern in all the buildings that house works of art.
4.
I think that is just a reminder of an old story.
https://www.getty.edu/conservation/publications_resources/newsletters/29_2/climate_effects.html
Although, the authors present their statement in a satisfactory way.
5.
Perhaps to compare an existing, no refurbished historical building and a refurbished one. Or, monitoring the Temp and RH, before and after some control strategies, for example by adding blinds to the windows.
Author Response
Response to Reviewer 2 Comments
Point 1: L. 678 “This result suggests the need of installing an air-conditioning system for the proper conservation of the artworks” but energy costs. Hence, the authors proposed to apply a different, more variation-tolerant approach, according to the standard EN 1575:2010, based “on the analysis of the historical climate within the exhibition zones.”
Point 2: Indeed, nothing new. For some examples, the fourth one is written by the same authors
1) Camuffo, D. Microclimate for cultural heritage; Elsevier: 1998.
2) Loupa, G.; Charpantidou, E.; Kioutsioukis, I.; Rapsomanikis, S. Indoor microclimate, ozone and nitrogen oxides in two medieval churches in Cyprus. Atmospheric Environment 2006, 40, 7457-7466, doi:10.1016/j.atmosenv.2006.07.015.
3) Corgnati, S.P.; Fabi, V.; Filippi, M. A methodology for microclimatic quality evaluation in museums: Application to a temporary exhibit. Building and Environment 2009, 44, 1253-1260, doi:https://doi.org/10.1016/j.buildenv.2008.09.012.
4) Sciurpi, F.; Carletti, C.; Cellai, G.; Pierangioli, L. Environmental monitoring and microclimatic control strategies in “La Specola” museum of Florence. Energy and Buildings 2015, 95, 190-201, doi:https://doi.org/10.1016/j.enbuild.2014.10.061.
5) Schito, E.; Testi, D.; Grassi, W. A Proposal for New Microclimate Indexes for the Evaluation of Indoor Air Quality in Museums. Buildings 2016, 6, 41.
Point 3: Also, L. 366-367 “This index allows the assessment of the quality of the indoor environment in relation to the conservation capability of the object exhibited inside the zone”.
This in not IAQ, is only microclimatic conditions. Aerosol is a major concern in all the buildings that house works of art.
Point 4: I think that is just a reminder of an old story.
https://www.getty.edu/conservation/publications_resources/newsletters/29_2/climate_effects.html
Although, the authors present their statement in a satisfactory way.
Point 5: Perhaps to compare an existing, no refurbished historical building and a refurbished one. Or, monitoring the Temp and RH, before and after some control strategies, for example by adding blinds to the windows.
Response: The authors are glad that the reviewer appreciated their work and thank for the comments. Based on these comments, careful modifications have been made to the manuscript. All changes are tracked in the revised version of the manuscript. In detail, we agree with reviewer’s point 3. The sentence L. 366-367 was revised by replacing “the quality of the indoor environment” with “the indoor microclimatic conditions”. Moreover, possible future developments in this research field are discussed in the Conclusions section according to the reviewer’s point 5.